# Semi-Supervised Classifier Guidance with Self-Calibration for Conditional Score-Based Generation

## Abstract

Score-based generative models (SGMs) are a popular family of deep generative models that achieve leading image generation quality. Early studies extend SGMs to tackle class-conditional generation by coupling an unconditional SGM with the guidance of a trained classifier. Nevertheless, such classifier-guided SGMs do not always achieve accurate conditional generation, especially when trained with fewer labeled data. We argue that the problem is rooted in the classifier's tendency to overfit without coordinating with the underlying unconditional distribution. To make the classifier respect the unconditional distribution, we propose improving classifier-guided SGMs by letting the classifier regularize itself. The key idea of our proposed method is to use principles from energy-based models to convert the classifier into another view of the unconditional SGM. Existing losses for unconditional SGMs can then be leveraged to achieve regularization by calibrating the classifier's internal unconditional scores. The regularization scheme can be applied to not only the labeled data but also unlabeled ones to further improve the classifier. Across various percentages of fewer labeled data, empirical results show that the proposed approach significantly enhances conditional generation quality. The enhancements confirm the potential of the proposed self-calibration technique for generative modeling with limited labeled data.

## 1 Introduction

Score-based generative models (SGMs) capture the underlying data distribution by learning the gradient function of the log-likelihood on data, also known as the score function. SGMs, when coupled with a diffusion process that gradually converts noise to data, can often synthesize higher-quality images than other popular alternatives, such as generative adversarial networks (Song et al., 2021; Dhariwal & Nichol, 2021). The community's research dedication to SGMs demonstrates promising performance in image generation (Song et al., 2021) and other fields such as audio synthesis (Kong et al., 2021; Jeong et al., 2021; Huang et al., 2022) and natural language generation (Li et al., 2022).

Many such successful SGMs focus on unconditional generation, which models the distribution without considering other variables (Song & Ermon, 2019; Ho et al., 2020; Song et al., 2021). When seeking to generate images controllably from a particular class, it is necessary to model the conditional distribution concerning another variable. Such *conditional* SGMs (Song et al., 2021; Dhariwal & Nichol, 2021; Chao et al., 2022) will be the focus of this paper.

There are two major families of conditional SGMs. Classifier-Free SGMs (CFSGMs) adopt specific conditional network architectures and losses (Dhariwal & Nichol, 2021; Ho & Salimans, 2021). CFSGMs are known to generate high-fidelity images when there are enough labeled data. Nevertheless, our findings indicate that their performance drops significantly as the proportion of labeled data decreases. That is, they have not met the needs of *semi-supervised* conditional generation with fewer labeled data. Classifier-Guided SGMs (CGSGMs) form another family of conditional SGMs (Song et al., 2021; Dhariwal & Nichol, 2021) based on decomposing the conditional score into the unconditional score plus the gradient of an auxiliary classifier with respect to the input. A vanilla CGSGM can then be constructed by learning a classifier in parallel to training an unconditional SGM with the popular Denoising Score Matching (DSM; Vincent, 2011) technique.

Because the unconditional SGM can be trained with both labeled and unlabeled data in principle, CGSGMs emerge with more potential than CFSGMs for the semi-supervised setting with fewer labeled data. The setting, which represents a realistic scenario when obtaining class labels is costly, will be the center of our study.

The quality of the classifier's gradients is critical for CGSGMs. If the classifier overfits (Lee et al., 2018; Müller et al., 2019; Mukhoti et al., 2020; Grathwohl et al., 2020) and produces highly inaccurate gradients, the resulting conditional scores may be unreliable. The unreliable conditional scores lower the generation quality even if the reliable unconditional scores can ensure decent generation fidelity. Although there are general regularization techniques (Zhang et al., 2019; Müller et al., 2019; Hoffman et al., 2019) that mitigate overfitting, their specific benefits for CGSGMs have not been fully studied. In fact, we find that those techniques are often not aligned with the unconditional SGM's view of the underlying distribution and offer limited benefits for improving CGSGMs. One pioneering enhancement of CGSGM on distribution alignment, Denoising Likelihood Score Matching (CG-DLSM; Chao et al., 2022), calibrates the classifier with a regularization loss that aligns the classifier's gradients to the ground-truth gradients with the *external* help of unconditional SGMs. Despite being able to achieve state-of-the-art performance within the CGSGM family, CG-DLSM is only designed for labeled data and cannot directly utilize the many unlabeled data in the semi-supervised setting.

In this work, we design a regularizer that calibrates the classifier *internally*, without any external help from the unconditional SGM. Such an internal regularization has been previously achieved by the Joint Energy-based Model (JEM; Grathwohl et al., 2020), which interprets classifiers as energy-based models. The interpretation allows JEM to define an auxiliary loss term that respects the underlying distribution and unlock the generation capability of classifiers when using MCMC sampling. Nevertheless, our careful study reveals that extending JEM as CGSGMs is highly non-trivial, as MCMC sampling is time-consuming and results in unstable loss values when coupled with the diffusion process. We thus choose not to extend JEM directly, and instead take inspiration from JEM to derive a novel CGSGM regularizer.

Our design broadens the JEM interpretation of the classifier. In particular, we show that the classifier, in addition to being an energy-based model (as JEM indicates), has a novel interpretation of being an unconditional SGM. Then, similar to how an external SGM is trained, we can regularize the classifier through its *internal* SGM interpretation, as illustrated with the Self-Calibration (SC) loss $\mathcal{L}_{\mathrm{SC}}$ in Fig. 2. The SC loss is efficient to compute and inherits sound theoretical guarantee and practical stability from the DSM technique for training unconditional SGMs. Our proposed CGSGM-SC approach trains the classifier component in Fig. 1 and the unconditional SGM *separately* to achieve efficiency. The approach applies the SC loss on both labeled and *unlabeled* data, resulting in immediate advantages in the semi-supervised setting.

Following earlier studies on CGSGMs (Chao et al., 2022), we visually study the effectiveness of CGSGM-SC on a synthesized data set. The results reveal that

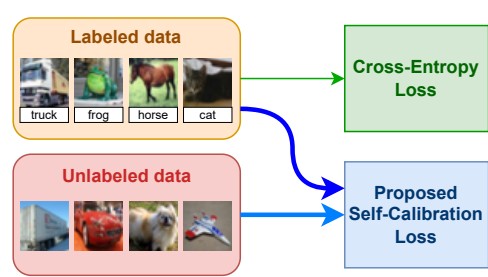

Figure 1: **Illustration of the proposed approach.** A vanilla CGSGM takes the green arrow (CE loss) to train the classifiers. The proposed CGSGM-SC additionally considers the two blue arrows representing the proposed self-calibration loss on both labeled and unlabeled data.

CGSGM-SC leads to more accurate classifier gradients than vanilla CGSGM, thus enhancing the estimation of conditional scores. We further conduct thorough experiments on standard benchmark data used by contemporary works (Chao et al., 2022) to validate the advantages of CGSGM-SC. The results demonstrate that CGSGM-SC is superior to the vanilla CGSGM and the state-of-the-art CGSGM-DLSM approach. Furthermore, in an extreme setting where only 5% of the data is labeled, CGSGM-SC is significantly better than all CGSGMs and CFSGMs by utilizes unlabeled data more effectively. The proposed method also brings additional benefits such as enabling classifier-only score estimation and improving adversarial robustness of

classifiers. The strong performance confirms the potential of CGSGM-SC in semi-supervised scenarios where labeled data are costly to obtain.

## 2 Background

Consider a data distribution $p(\boldsymbol{x})$ where $\boldsymbol{x} \in \mathbb{R}^d$. The purpose of an SGM is to generate samples from $p(\boldsymbol{x})$ via the information contained in the score function $\nabla_{\boldsymbol{x}} \log p(\boldsymbol{x})$, which is learned from data. We first introduce how a diffusion process can be combined with learning a score function to effectively sample from $p(\boldsymbol{x})$ in Section 2.1. Next, a comprehensive review of works that have extended SGMs to conditional SGMs is presented in Section 2.2, including those that incorporates classifier regularization for CGSGMs. Finally, JEM (Grathwohl et al., 2020) is introduced in Section 2.3, highlighting its role in inspiring our proposed methodology.

### 2.1 Score-based generative modeling by diffusion

Song et al. (2021) propose to model the transition from a known prior distribution $p_T(\boldsymbol{x})$, typically a multivariate gaussian noise, to an unknown target distribution $p_0(\boldsymbol{x}) = p(\boldsymbol{x})$ using the Markov chain described by the following stochastic differential equation (SDE):

$$d\boldsymbol{x} = \left[f(\boldsymbol{x}, t) - g(t)^2 s(\boldsymbol{x}, t)\right] dt + g(t) d\bar{\boldsymbol{w}}, \tag{1}$$

where $\bar{\boldsymbol{w}}$ is a standard Wiener process when the timestep flows from $T$ back to 0, $s(\boldsymbol{x}, t) = \nabla_{\boldsymbol{x}} \log p_t(\boldsymbol{x})$ denotes a time-dependent score function, and $f(\boldsymbol{x}, t)$ and $g(t)$ are prespecified functions that describe the overall movement of $p_t(\boldsymbol{x})$, the noise-disturbed distribution until time $t$. The score function is learned by optimizing the following time-generalized Denoise Score Matching (DSM; Vincent, 2011) loss

$$\mathcal{L}_{DSM}(\boldsymbol{\theta}) = \mathbb{E}_t \left[\lambda_t \mathbb{E}_{\boldsymbol{x}_t, \boldsymbol{x}_0} \left[\frac{1}{2} \|s(\boldsymbol{x}_t, t; \boldsymbol{\theta}) - s_t(\boldsymbol{x}_t | \boldsymbol{x}_0)\|_2^2\right]\right], \tag{2}$$

where $t$ is selected uniformly between 0 and $T$, $\boldsymbol{x}_t \sim p_t(\boldsymbol{x})$, $\boldsymbol{x}_0 \sim p_0(\boldsymbol{x})$, $s_t(\boldsymbol{x}_t | \boldsymbol{x}_0)$ denotes the score function of the noise distribution $p_t(\boldsymbol{x}_t | \boldsymbol{x}_0)$, which can be calculated using $f(\boldsymbol{x}, t)$ and $g(t)$, and $\lambda_t$ is a weighting function that balances the loss of different timesteps. In this paper, we use the hyperparameters from the original VE-SDE framework (Song et al., 2021). A more detailed introduction to learning the score function and sampling through SDEs is described in Appendix C.

### 2.2 Conditional score-based generative models

In the semi-supervised setting for learning conditional SGMs, we are given labeled data $\{(\boldsymbol{x}_m, y_m)\}_{m=1}^M$ in addition to unlabeled data $\{\boldsymbol{x}_n\}_{n=M+1}^{M+N}$, where $y_m \in \{1, 2, \ldots, K\}$ denotes the class label. Here the number of unlabeled data $N$ is typically much larger than the number of labeled data $M$, which represents the realistic scenario with abundant unlabeled data. The goal of conditional SGMs is to learn the conditional score function $\nabla_{\boldsymbol{x}} \log p(\boldsymbol{x}|y)$ and then generate samples from $p(\boldsymbol{x}|y)$, typically using a diffusion process as discussed in Section 2.1 and Appendix C.2.

One approach for conditional SGMs is CFSGM (Dhariwal & Nichol, 2021; Ho & Salimans, 2021), which parameterizes its model with a joint architecture such that the class labels $y$ can be included as inputs. Classifier-Free Guidance (CFG; Ho & Salimans, 2021) is a well-known representative of CFSGM that additionally uses a null token $y_{\text{NIL}}$ to indicate unconditional score calculation. The unconditional score is linearly combined with the conditional score calculation for some specific $y$ to form the final estimate of $s(\boldsymbol{x}|y)$. CFG is a state-of-the-art conditional SGM when $N = 0$ and $M$ is sufficiently large—the fully-supervised setting. Nevertheless, as we shall show in our experiments, its performance drops significantly in the semi-supervised setting, as there may not be enough labeled data for training the conditional parts of CFG.

---

**Algorithm 1** Semi-supervised classifier training in vanilla classifier-guidance

---

**Input:** Labeled data $D_l$
Initialize the time-dependent classifier $f(\boldsymbol{x}, y, t; \boldsymbol{\phi})$ randomly
**repeat**
    Sample data $(\boldsymbol{x}_l, y_l) \sim D_l$
    Sample timesteps $t_l \sim \text{Uniform}(1, T)$
    Obtain perturbed data $\tilde{\boldsymbol{x}}_l \sim p_{t_l}(\boldsymbol{x}|\boldsymbol{x}_l)$
    Calculate $\mathcal{L}_{CE} = \text{CrossEntropy}(f(\boldsymbol{x}_l, y, t; \boldsymbol{\phi}), y_l)$
    Take gradient step on $\mathcal{L}_{CE}$
**until** converged

---

Another popular family of conditional SGM is CGSGM, which decomposes the conditional score function using Bayes' theorem (Song et al., 2021; Dhariwal & Nichol, 2021):

$$\nabla_{\boldsymbol{x}} \log p(\boldsymbol{x}|y) = \nabla_{\boldsymbol{x}}[\log p(\boldsymbol{x}) + \log p(y|\boldsymbol{x}) - \log p(y)]$$
$$= \nabla_{\boldsymbol{x}} \log p(\boldsymbol{x}) + \nabla_{\boldsymbol{x}} \log p(y|\boldsymbol{x}) \tag{3}$$

The $\log p(y)$ term can be dropped because it is not a function of $\boldsymbol{x}$ and is thus of gradient 0. The decomposition shows that conditional generation can be achieved by an unconditional SGM that learns the score function $\nabla_{\boldsymbol{x}} \log p(\boldsymbol{x})$ plus an extra conditional gradient term $\nabla_{\boldsymbol{x}} \log p(y|\boldsymbol{x})$ with respect to the input $\boldsymbol{x}$.

The vanilla Classifier-Guidance (CG) approach in the CGSGM family estimates $\nabla_{\boldsymbol{x}} \log p(y|\boldsymbol{x})$ with an auxiliary classifier trained from the cross-entropy loss on the labeled data and learns the unconditional score function by the denoising score matching loss $\mathcal{L}_{\text{DSM}}$, which in principle can be applied on both unlabeled and labeled data. Algorithm 1 summarizes the training of such time-dependent classifiers in semi-supervised scenarios. Nevertheless, the classifier within the vanilla CG approach is known to be potentially overconfident (Lee et al., 2018; Müller et al., 2019; Mukhoti et al., 2020; Grathwohl et al., 2020) in its predictions, which in turn results in inaccurate gradients. This can mislead the conditional generation process and decrease class-conditional generation quality.

Dhariwal & Nichol (2021) propose a heuristic to address the over-confidence issue by post-processing the term $\nabla_{\boldsymbol{x}} \log p(y|\boldsymbol{x})$ with a scaling parameter $\lambda_{CG} \neq 1$.

$$\nabla_{\boldsymbol{x}} \log p(\boldsymbol{x}|y) = s(\boldsymbol{x}) + \lambda_{CG} \nabla_{\boldsymbol{x}} \log p(y|\boldsymbol{x}), \tag{4}$$

where $s(\boldsymbol{x}) = \nabla_{\boldsymbol{x}} \log p(\boldsymbol{x})$ is the unconditional score and $p(y|\boldsymbol{x})$ is the classifier. Decreasing $\lambda_{CG}$ effectively smoothens the classifier, guiding the generation process to produce more diverse but lower fidelity samples. While the tuning heuristic is effective in exploring the diversity-fidelity trade-off and improving the vanilla CG, it is not backed by sound theoretical explanations.

A promising research direction is to regularize the classifier *during training* for resolving the overconfidence issue. For instance, CGSGM with Denoising Likelihood Score Matching (CG-DLSM; Chao et al., 2022) presents a regularization technique that employs the DLSM loss below formulated from the classifier gradient $\nabla_{\boldsymbol{x}} \log p(y, t|\boldsymbol{x}; \boldsymbol{\phi})$ and the unconditional score function $s_t(\boldsymbol{x})$.

$$\mathcal{L}_{\text{DLSM}}(\boldsymbol{\phi}) = \mathbb{E}_t \left[ \lambda_t \mathbb{E}_{\boldsymbol{x}_t, \boldsymbol{x}_0} \left[ \frac{1}{2} \left\| \nabla_{\boldsymbol{x}} \log p_t(y|\boldsymbol{x}_t; \boldsymbol{\phi}) + s_t(\boldsymbol{x}_t) - s_t(\boldsymbol{x}_t|\boldsymbol{x}_0) \right\|_2^2 \right] \right], \tag{5}$$

where $\boldsymbol{\phi}$ represents the classifier's parameters, and the unconditional score function $s_t(\boldsymbol{x})$ is estimated via an unconditional SGM $s(\boldsymbol{x}_t, t; \boldsymbol{\theta})$. The CG-DLSM authors (Chao et al., 2022) prove that Eq. 5 can calibrate the classifier to produce more accurate gradients $\nabla_x \log p(y|\boldsymbol{x})$.

## 2.3 Reinterpreting classifiers as energy-based models

Our proposed methodology draws inspiration from JEM (Grathwohl et al., 2020), which shows that reinterpreting classifiers as energy-based models (EBMs) and enforcing regularization with related objectives

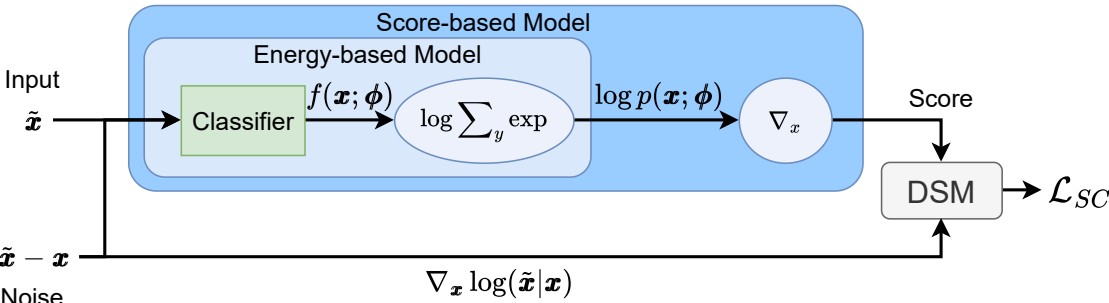

Figure 2: **Calculation of the proposed self-calibration loss.** First, perturbed sample $\tilde{x}$ is fed to the time-dependent classifier to obtain the output logits. Second, the logits are transformed into log-likelihood by applying $\log \sum_y \exp$. Third, we calculate its gradient w.r.t. the input $\tilde{x}$ to obtain the estimated score. Finally, denoising score matching is applied to obtain the proposed self-calibration loss. The proposed loss is used as an auxiliary loss to train the classifier.

helps classifiers to capture more accurate probability distributions. EBMs are models that estimate energy functions $E(\boldsymbol{x})$ of distributions (LeCun et al., 2006), which satisfies $\log p(\boldsymbol{x}) = -E(\boldsymbol{x}) + \log \int_{\boldsymbol{x}} \exp(E(\boldsymbol{x}))d\boldsymbol{x}$. Given the logits of a classifier to be $f(\boldsymbol{x}, y; \boldsymbol{\phi})$, the estimated joint distribution can be written as

$$p(\boldsymbol{x}, y; \boldsymbol{\phi}) = \frac{\exp(f(\boldsymbol{x}, y; \boldsymbol{\phi}))}{Z(\boldsymbol{\phi})},$$

where $\exp(\cdot)$ means exponential and $Z(\boldsymbol{\phi}) = \int_{\boldsymbol{x}, y} \exp(f(\boldsymbol{x}, y; \boldsymbol{\phi})) \, d\boldsymbol{x} \, dy$. After that, the energy function $E(\boldsymbol{x}; \boldsymbol{\phi})$ can be obtained by

$$E(\boldsymbol{x}; \boldsymbol{\phi}) = -\log \Sigma_y \exp(f(\boldsymbol{x}, y; \boldsymbol{\phi})) \tag{6}$$

Then, losses used to train EBMs can be seamlessly leveraged in JEM to regularize the classifier, such as the typical EBM loss $\mathcal{L}_{\text{EBM}} = \mathbb{E}_{p(\boldsymbol{x})} [-\log p(\boldsymbol{x}; \boldsymbol{\phi})]$. JEM uses MCMC sampling for computing the loss and is shown to result in a well-calibrated classifier in their empirical study. The original JEM work (Grathwohl et al., 2020) also reveals that classifiers can be used as a reasonable generative model, but its generation performance is knowingly worse than state-of-the-art SGMs.

## 3    The proposed self-calibration methodology

In this work, we consider CGSGMs under the diffusion generation process as discussed in Section 2.1. Such CGSGMs require learning an unconditional SGM, which is assumed to be trained with denoising score matching (DSM; Vincent, 2011) due to its close relationship with the diffusion process. Such CGSGMs also require a time-dependent classifier that models $p_t(y|\boldsymbol{x})$ instead of $p(y|\boldsymbol{x})$, which can be done by applying a time-generalized cross-entropy loss.

Section 2.2 has illustrated the importance of regularizing the classifier to prevent it from misguiding the conditional generation process. One naive thought is to use established regularization techniques, such as label-smoothing and Jacobian regularization (Hoffman et al., 2019). Those techniques that are less attached to the underlying distribution will be studied in Section 4. Our proposed regularization loss, inspired by the success of DLSM Chao et al. (2022) and JEM (Grathwohl et al., 2020), attempts to connect with the underlying distribution better. Moreover, the proposed self-calibration technique could be seamlessly applied in the semi-supervised settings by utilizing unlabeled data for regularization, which will be further discussed in Section 3.3.

### 3.1    Formulation of self-calibration loss

We extend JEM (Grathwohl et al., 2020) to connect the time-dependent classifier to the underlying distribution. In particular, we reinterpret the classifier as a time-dependent EBM. The interpretation allows us to

obtain a time-dependent version of $p_t(\boldsymbol{x})$ within the classifier, which can be used to obtain a classifier-*internal* version of the score function. Then, we propose to regularize the classifiers with a loss on their internal score functions $\nabla_{\boldsymbol{x}} \log p_t(\boldsymbol{x})$ instead of regularizing them by the EBM loss $-\log p_t(\boldsymbol{x})$ like JEM.

Under the JEM interpretation, the time-dependent energy function is $E(\boldsymbol{x}, t; \boldsymbol{\phi}) = -\log \Sigma_y \exp(f(\boldsymbol{x}, y, t; \boldsymbol{\phi}))$, where $f(\boldsymbol{x}, y, t; \boldsymbol{\phi})$ is the output logits of the time-dependent classifier. Then, the internal time-dependent unconditional score function is

$$s^c(\boldsymbol{x}, t; \boldsymbol{\phi}) = \nabla_{\boldsymbol{x}} \log \Sigma_y \exp(f(\boldsymbol{x}, y, t; \boldsymbol{\phi})), \tag{7}$$

where $s^c$ is used instead of $s$ to indicate that the unconditional score is computed *within* the classifier. Then, we adopt the standard DSM technique in Eq. 2 to "train" the internal score function, forcing it to follow its physical meaning during the diffusion process. The resulting self-calibration loss can then be defined as

$$\mathcal{L}_{\text{SC}}(\boldsymbol{\phi}) = \mathbb{E}_t \left[ \lambda_t \mathbb{E}_{\boldsymbol{x}_t, \boldsymbol{x}_0} \left[ \frac{1}{2} \left\| s^c(\boldsymbol{x}_t, t; \boldsymbol{\phi}) - s_t(\boldsymbol{x}_t | \boldsymbol{x}_0) \right\|_2^2 \right] \right] \tag{8}$$

Similar to Eq. 2, $t$ is selected uniformly between 0 and $T$, $\boldsymbol{x}_t \sim p_t(\boldsymbol{x})$, $\boldsymbol{x}_0 \sim p_0(\boldsymbol{x})$, $s_t(\boldsymbol{x}_t | \boldsymbol{x}_0)$ denotes the score function of the noise distribution $p_t(\boldsymbol{x}_t | \boldsymbol{x}_0)$, which can be calculated using $f(\boldsymbol{x}, t)$ and $g(t)$, and $\lambda_t$ is a weighting function that balances the loss of different timesteps. Fig. 2 summarizes the calculation of the proposed SC loss. In this work, we adopt VE-SDE (Song et al., 2021) as the backbone for score-based generative modeling.

After the self-calibration loss is obtained, it is mixed with the cross-entropy loss $\mathcal{L}_{\text{CE}}$ to train the classifier. The total loss can be written as:

$$\mathcal{L}_{\text{CLS}}(\boldsymbol{\phi}) = \mathcal{L}_{\text{CE}}(\boldsymbol{\phi}) + \lambda_{SC} \mathcal{L}_{\text{SC}}(\boldsymbol{\phi}), \tag{9}$$

where $\lambda_{SC}$ is a tunable hyper-parameter. The purpose of self-calibration is to guide the classifier to estimate the score function of the underlying data distribution more accurately, which in term leads to a better-regularized classifier that respects the the underlying distribution. After self-calibration, the classifier is then used in CGSGM to guide an unconditional SGM for conditional generation. Note that since our approach regularizes the classifier during training while classifier gradient scaling (Eq. 4) is done during sampling, we can easily combine the two techniques to enhance performance. We will demonstrate the possibility in Section 3.4.

### 3.2 Comparison with related regularization methods

This section provides a comparative analysis of the regularization methods like label-smoothing and Jacobian regularization (Hoffman et al., 2019) and the ones employed in DLSM (Chao et al., 2022), JEM Grathwohl et al. (2020), and our proposed self-calibration loss.

**Label-smoothing and Jacobian regularization (Hoffman et al., 2019)** These methods aim to regularize classifiers by incorporating loss functions based on assumptions about the underlying distribution. Although these techniques have been shown to improve robustness, they do not align well with the view of the underlying distribution inherent in unconditional SGMs.

**DLSM (Chao et al., 2022)** DLSM and our proposed method both regularize the classifier to align better with unconditional SGMs' view of the underlying distribution. DLSM achieves this by relying on the help of an *external* trained SGM, whereas self-calibration regularizes the classifier by using a classifier-*internal* SGM. Furthermore, the design of DLSM loss can only utilize labeled data, while our method is able to make use of all data.

**JEM (Grathwohl et al., 2020)** JEM interprets classifiers as unconditional EBMs, and self-calibration further extends the interpretation to unconditional SGMs. The training stage of EBM that incorporates MCMC sampling is known to be unstable and time-consuming. Even though one can possibly mitigate the instability issue by increased sampling steps and additional hyperparameter tuning, doing so largely lengthens the training time. Our proposed self-calibration precludes the need for MCMC sampling during training and substantially improves both stability and efficiency in our experiments.

---

**Algorithm 2** Semi-supervised classifier training with self-calibration loss

---

**Input:** Labeled data $D_l$, unlabeled data $D_u$
Initialize the time-dependent classifier $f(\boldsymbol{x}, y, t; \boldsymbol{\phi})$ randomly
**repeat**
    Sample data $(\boldsymbol{x}_l, y_l) \sim D_l$, $\boldsymbol{x}_u \sim D_u$
    Sample timesteps $t_l, t_u \sim \text{Uniform}(1, T)$
    Obtain perturbed data $\tilde{\boldsymbol{x}}_l \sim p_{t_l}(\boldsymbol{x}|\boldsymbol{x}_l), \tilde{\boldsymbol{x}}_u \sim p_{t_u}(\boldsymbol{x}|\boldsymbol{x}_u)$
    Calculate $\mathcal{L}_{CE} = \text{CrossEntropy}(f(\boldsymbol{x}_l, y, t; \boldsymbol{\phi}), y_l)$
    # Use the same amount of labeled and unlabeled data to calculate the self-calibration loss
    Calculate $\mathcal{L}_{SC} = \mathbb{E}_{(\boldsymbol{x}, t) \in \{(\boldsymbol{x}_l, t_l), (\boldsymbol{x}_u, t_u)\}} \left[ \frac{1}{2} \lambda_t \| \nabla_{\boldsymbol{x}} \log \Sigma_y \exp(f(\boldsymbol{x}, y, t; \boldsymbol{\phi})) - s_t(\boldsymbol{x}_t|\boldsymbol{x}_0) \|_2^2 \right]$
    Take gradient step on $\mathcal{L}_{CLS} = \mathcal{L}_{CE} + \lambda_{SC} \mathcal{L}_{SC}$
**until** converged

---

**CG-JEM** In contrast to the previous paragraph, this paragraph discusses the possibility of directly incorporating JEM into the time-dependent CGSGM framework. In our experiments, JEM guidance (CG-JEM) performs around the same as vanilla classifier guidance but possesses several significant disadvantages. Coupling EBM training with additional training objectives is known to introduce increased instability, especially for time-dependent classifiers, considering it is more difficult to generate meaningful time-dependent data through MCMC sampling. Such instability makes it hard for the training losses to converge to a low and steady stage. While we can increase the MCMC sampling step to enhance the stability of loss calculation, doing so poses prohibitive computational costs to the training process. For example, in our naive implementation of time-dependent JEM, it either (1) incurs high instability (loss diverges within $10,000$ steps in all 10 runs, and requiring 4.23s per step) or (2) poses unaffordable resource requirements (20s per step, approximately 50 days in total). While enhanced stability can be achieved after incorporating diffusion recovery likelihood (Gao et al., 2021), the time-costly nature of MCMC sampling remains (6.03s per step). In contrast, our proposed method circumvents the need for MCMC sampling during training, significantly improving both stability and efficiency by requiring only 0.75s per step.

**CG-SC (Ours)** The proposed method utilizes the score function estimated by the classifier for regularization. This direct calibration of the classifier-estimated score function offers significant advantages over existing methods. Unlike DLSM, which depends on an external trained SGM and can only leverage labeled data, our approach can utilize all available data and provide more benefits in semi-supervised scenarios. Furthermore, compared to JEM and CG-JEM, which requires unstable and time-consuming MCMC sampling, self-calibration avoids such complexities, greatly enhancing both stability and training efficiency. Overall, CG-SC presents a robust, efficient, and theoretically sound method for classifier regularization, making it a superior alternative to the aforementioned methods.

### 3.3 Self-calibration for semi-supervised learning

The lack of labels in the semi-supervised setting constitutes a great challenge to learning an accurate classifier. The SC loss derived in Section 3 can be natively applied for the semi-supervised setting by considering both labeled and unlabeled data. To incorporate the loss and utilize unlabeled data during training, we change the way $\mathcal{L}_{\text{CLS}}$ is calculated from Eq. 9 by using only labeled data to calculate the cross-entropy loss ($\mathcal{L}_{CE}$) and all available data to calculate the proposed self-calibration loss ($\mathcal{L}_{SC}$). As illustrated in Fig. 1, both labeled and unlabeled data could be used to calculate $\mathcal{L}_{\text{SC}}$, whereas $\mathcal{L}_{\text{CE}}$ could only be applied to labeled data.

During training, we observe that when there is more than 80% of unlabeled data, the cross-entropy loss does not converge to a low-and-steady stage if the algorithm randomly samples from all training data. We conjecture that the slow convergence may be due to the low percentage of labeled data within each mini-batch during training. Therefore, we propose to change the way we sample batches by always ensuring that exactly half of the data is labeled. Algorithm 2 summarizes the semi-supervised training process of the classifier.

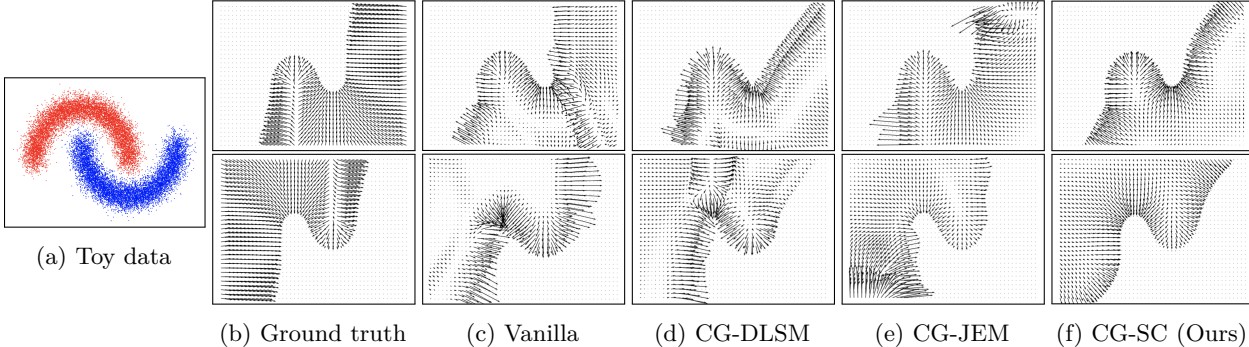

(a) Toy data

(b) Ground truth     (c) Vanilla     (d) CG-DLSM     (e) CG-JEM     (f) CG-SC (Ours)

Figure 3: **Gradients of classifiers $\nabla_{\boldsymbol{x}} \log p(y|\boldsymbol{x})$ for toy dataset.** The upper row contains the gradients for class 1 (red), and the lower contains the gradients for class 2 (blue). (a) Real data distribution. (b) Ground truth classifier gradients. Gradients estimated by (c) Vanilla CG, (d) CG-DLSM, (e) CG-JEM, and (f) CG with proposed self-calibration. We observed that the gradients estimated by the vanilla classifier are highly inaccurate and fluctuate greatly. On the other hand, regularized classifiers produce gradients that are closer to the ground truth and contain fewer fluctuations.

## 3.4 Demonstration experiments on a 2D toy dataset

Following DLSM (Chao et al., 2022), we use a 2D toy dataset containing two classes to demonstrate the effects of self-calibration loss and visualize the training results. The data distribution is shown in Fig. 3a, where the two classes are shown in two different colors. After training the classifiers on the toy dataset with different methods, we plot the gradients $\nabla_{\boldsymbol{x}} \log p(y|\boldsymbol{x})$ at the minimum timestep $t = 0$ estimated by the classifiers and compare them with the ground truth.

Fig. 3 shows the ground truth classifier gradient and the gradients estimated by different classifiers. The Vanilla CGSGM, which takes an unregularized classifier, produces gradients that rapidly change in magnitude across the 2D space, with frequent fluctuations and mismatches with the ground truth. Such fluctuations can impede the convergence of the reverse diffusion process to a stable data point, leading to noisier samples. Moreover, the divergence from the ground truth gradient can misguide the SGM, leading to generation of samples from incorrect classes. The unregularized classifier within Vanilla CGSGM also tends to generate large gradients near the distribution borders and tiny gradients elsewhere. This implies that when the sampling process is heading toward the incorrect class, such classifiers are not able to "guide" the sampling process back towards the desired class. In comparison, the introduction of various regularization techniques such as DLSM, JEM, and the proposed self-calibration technique results in estimated gradients that are more stable, continuous across the 2D space, and better aligned with the ground truth. This stability brings about a smoother generation process and the production of higher-quality samples.

Table 1 shows the quantitative measurements of the methods on the toy dataset. First, we compare the gradients $\nabla_{\boldsymbol{x}} \log p(y|\boldsymbol{x})$ estimated by the classifiers with the ground truth by calculating the mean squared error (first column) and cosine similarity (second column). The results were calculated by averaging over all $(x, y) \in \{(x, y): x \in \{-12, -11.5, \ldots, 11.5, 12\}, y \in \{-8, -7.5, \ldots, 7.5, 8.0\}\}$. We observe that after self-calibration, the mean squared error of the estimated gradients is 18% lower compared to Vanilla CGSGM. In addition, taking scaling heuristic in Eq. 4 for post-processing further improves the gradient estimation error by 36%. This improvement after scaling implies that the direction of gradients predicted by self-calibrated classifiers better aligns with the ground truth, and scaling further reduces the mismatch between the magnitude of the classifier and the ground truth. In terms of cosine similarity, incorporating self-calibration introduces an improvement of 42%. Although CG-JEM achieves the best gradient directions, it also leads to greater fluctuations in its gradient magnitude compared to the proposed CG-SC (Fig. 3). Such significant fluctuations result in worse mean squared error of gradients and cause scaling to provide minimal overall improvements, as the top-right and bottom-left parts require more scaling while other parts need less. The numerical results support our previous observation: after self-calibration, classifiers better align with the ground truth compared to all other methods.

Table 1: **Quantitative measurements of all CGSGM methods tested on the toy dataset.** MSE and CS stand for mean squared error and cosine similarity, respectively. The proposed self-calibration consistently outperforms the vanilla classifier. Furthermore, the self-calibrated classifier achieves the best mean squared error, and its cosine-similarity is also close to the best-performing CG-JEM.

| Method | Gradient MSE ($\downarrow$) | Gradient CS ($\uparrow$) | Cond-Score CS ($\uparrow$) |
|---|---|---|---|
| CG | 8.7664 | 0.3348 | 0.9175 |
| + scaling | 8.1916 | 0.3348 | 0.9447 |
| CG-DLSM | 8.1183 | 0.4450 | 0.9316 |
| + scaling | 8.0671 | 0.4450 | 0.9328 |
| CG-JEM | 8.5577 | **0.6429** | 0.9670 |
| + scaling | 8.5577 | **0.6429** | **0.9709** |
| CG-SC (Ours) | 7.1558 | 0.5758 | 0.9454 |
| + scaling | **5.6376** | 0.5758 | 0.9689 |

Besides comparing the gradients predicted by different classifiers, we also analyzed the conditional score predicted using the classifier guidance framework. We first added the unconditional score of the training data distribution to the classifier gradients to calculate the conditional scores and compared the results with the ground truth. We observe that vanilla CG estimates conditional scores with a cosine similarity of 0.9175, which shows that CGSGMs can produce reasonable conditional scores even with sub-optimal classifiers. This observation explains why vanilla CGSGMs generate samples with decent quality. After applying self-calibration loss and the scaling method, we further improve the cosine similarity to 0.9689, which enhances both the accuracy of conditional score estimation and the quality of class-conditional generation.

## 4 Experiments

In this section, we test our proposed CG-SC method and other relevant methods on the CIFAR-10 and CIFAR-100 datasets. We demonstrate that our method improves generation both conditionally and unconditionally with different percentages of labeled data. We also include in Appendix the test accuracies of different time-dependent classifiers trained on the CIFAR-10 dataset (Appendix E), the performance of CG-SC across various training steps (Appendix F), and randomly selected images of CGSGM before and after self-calibration on CIFAR-10 (Appendix G).

### 4.1 Experimental setup

**Evaluation metrics** We primarily focus on metrics that directly evaluate the quality of conditional generation. In this setting, each class introduces one or more additional modes, making intra-class evaluation far more informative than a single global metric. To this end, we report the following conditional metrics (computed per class then averaged across all classes) as well as their unconditional counterparts for completeness (see Table 2):

- **Intra-FID and FID** (Heusel et al., 2017): These metrics compare the generated distribution to the target distribution in the latent space of a pre-trained feature extractor, which is the Inception network in our experiments. Lower FID scores indicate higher similarity. While FID assesses the overall match between the aggregated distributions, intra-FID is more expressive in the conditional generation setting as it captures the similarity of distributions on a per-class basis, reflecting the model's ability to represent each class-conditional distribution.

- **Intra-Density and Density** (Naeem et al., 2020): These metrics measure the proportion of generated samples that fall within the in-distribution support of the target data. Higher density scores indicate that fewer samples are out-of-distribution. Similar to intra-FID, we view intra-Density as a more expressive metric in class-conditional generation. This is because generating samples to the

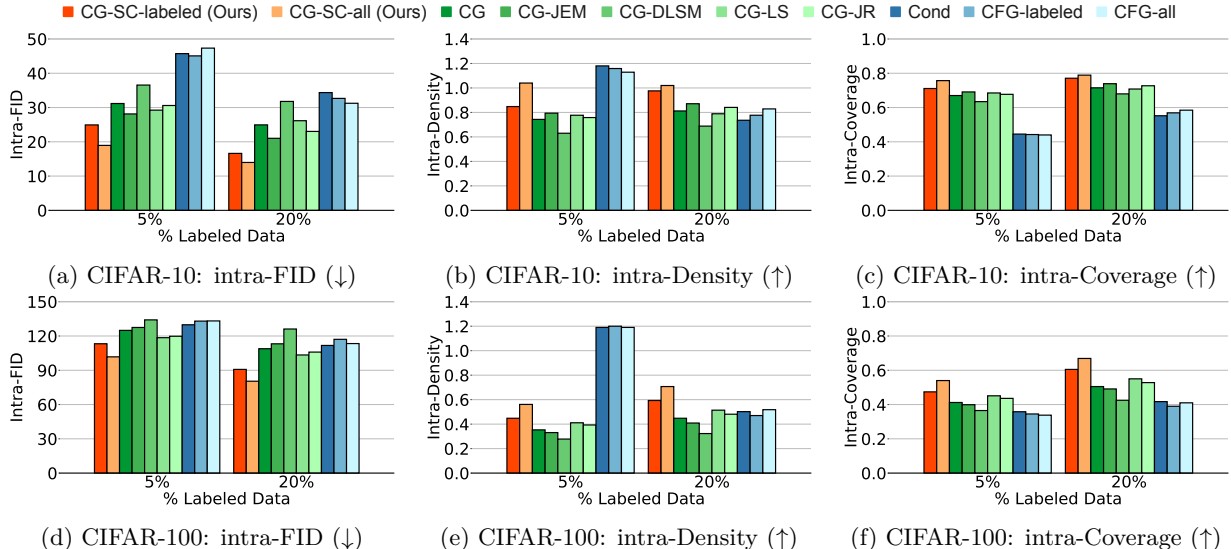

Figure 4: **Results of class-conditional generation in semi-supervised settings.** In semi-supervised settings, although CFSGMs generate images with high intra-Density, which demonstrates they are able to generate images to the right class, the images only cover a small portion of the class-conditional distributions. This makes CGSGMs preferable as fewer labeled data does not cause the generation diversity of CGSGMs to decrease by much. Compared to other CGSGMs, the proposed self-calibration consistently achieves the best conditional generative performance.

> incorrect class would not affect Density, as samples generated to the incorrect class are not out-of-distribution to unconditional distributions; in contrast, such inaccuracies would be reflected in intra-Density.

- **Intra-Coverage and coverage** (Naeem et al., 2020): These metrics evaluate how well the diversity of the generated distribution aligns with that of the target distribution. Higher coverage scores indicate better modeling of this diversity. While Coverage evaluates the overall capture of variations, Intra-Coverage is more expressive in assessing conditional generation as it implies the model's capacity to represent the full range of variations within each class.

In comparison their unconditional counterparts, the main advantage of intra-class metrics is the ability to reflect class-conditional generation accuracy. For instance, if we swap the samples in classes "airplane" and "automobile" of the training data, we would still obtain perfect FID, Density, and Coverage scores, whereas intra-fid, intra-density, and intra-coverage scores would significantly deteriorate. By focusing on these conditional metrics, we gain a more comprehensive understanding of class-conditional generative performance that goes beyond what unconditional metrics can reveal.

**Proposed methods**  In this section, we present two variants of the proposed method: CG-SC-labeled and CG-SC-all. **CG-SC-labeled** refers to the classifier guidance results where the classifier is regularized using the proposed self-calibration loss on labeled data, whereas **CG-SC-all** refers results where the classifier is regularized with the proposed self-calibration loss on both labeled and unlabeled data.

**Baseline methods (CGSGMs)**  The proposed methods are compared with the following variants of classifier-guided SGMs, including the vanilla CG and CG with the regularization methods discussed in Section 3.2. **CG**: vanilla classifier guidance; **CG-JEM**: classifier guidance with JEM loss; **CG-DLSM**: classifier guidance with DLSM loss (Chao et al., 2022); **CG-LS**: classifier guidance with label-smoothing; **CG-JR**: classifier guidance with Jacobian regularization (Hoffman et al., 2019).

**Baseline methods (CFSGMs)**  The proposed methods are compared with the following CFSGMs, covering major variations of CFSGMs in our experimental settings that do not utilize additional data. **Cond** trains conditional SGMs solely on labeled data, sampling directly from the conditional score; **CFG-labeled** trains both conditional and unconditional paths using only labeled data; **CFG-all** uses all data for training the unconditional path. While CFSGMs are the current state-of-the-art in fully supervised settings, our experiments show that they suffer from reduced diversity in semi-supervised scenarios due to overfitting on limited labeled data.

**Experimental details**  We implemented the unconditional score estimation model following the approach outlined in NCSN++ (Song et al., 2021). Additionally, we adapted the encoder component of NCSN++ as the classifier for CGSGM (Dhariwal & Nichol, 2021) and its variants, including CG-DLSM, the proposed CG-SC, and other CGSGM-based methods. For sampling, we employed Predictor-Corrector (PC) samplers (Song et al., 2021) with 1000 steps. The SDE was based on the VE-SDE framework proposed by Song et al. (2021). The hyper-parameter introduced in Eq. 9 was tuned within the range $\{10, 1, 0.1, 0.01\}$ for fully-supervised settings and fixed at 1 for semi-supervised settings due to limited computational resources. The scaling factor $\lambda_{CG}$ in Eq. 4 was optimized within $\{0.5, 0.8, 1.0, 1.2, 1.5, 2.0, 2.5\}$ to achieve the best intra-FID. Similarly, the scaling factor $\lambda_{CFG}$ for classifier-free SGMs was tuned within $\{0, 0.1, 0.2, 0.4\}$ to obtain optimal intra-FID. The balancing factors for the DLSM loss and the Jacobian regularization loss were set to 1 and 0.01, respectively, as recommended in the original literature. The label-smoothing factor was tuned between $\{0.1, 0.05\}$ for the better intra-FID.

## 4.2 Experimental results

Table 2 and Fig. 4 present the performance of all methods when applied to various percentages of labeled data.

**Classifier-free SGMs (CFSGMs) vs Classifier-guided SGMs (CGSGMs)**  The first observation from Fig. 4 is that CFSGMs, including Cond, CFG-labeled, and CFG-all, consistently excel in the intra-Density metric, demonstrating their ability to generate class-conditional images with high accuracy. However, Table 2 and Fig. 4 reveal a significant performance drop in terms of Coverage, intra-FID, and intra-Coverage when CFSGMs perform class-conditional generation tasks with few labeled data. While CFSGMs generate images with high quality and accuracy, they tend to lack diversity when working with fewer labeled data. This is primarily due to insufficient labeled data during training, causing them to generate samples that closely mirror the distribution of only the labeled data, rather than the entire data set. In other words, the conditional paths of CFSGMs tend to severely overfit on labeled data, causing the sample diversity to deteriorate when labeled data is scarce. This limitation deteriorates Coverage and intra-Coverage, leading to poor performance in terms of intra-FID as the generated distribution does not fully resemble the target distribution. In contrast, CGSGMs, which leverage both labeled and unlabeled data during training in semi-supervised settings, perform better in semi-supervised class-conditional generation. Moreover, the diversity measures, including coverage and intra-Coverage, of CGSGMs are more consistent across various percentages of labeled data (Table 2). This indicates that having less labeled data does not significantly reduce the diversity of generated images for CGSGMs, making CGSGMs advantageous over CFSGMs in semi-supervised scenarios.

**Regularized CGSGMs vs Vanilla CGSGM**  Basic regularization methods like label-smoothing and Jacobian regularization (Hoffman et al., 2019) show consistent but marginal improvement over vanilla CGSGM. This points out that although these methods mitigate overfitting on training data, the constraints they enforce do not align with SGMs, limiting the benefit of such methods. CG-DLSM (Chao et al., 2022), on the other hand, achieves great unconditional generation performance in all settings. However, its class-conditional performance suffers from a significant performance drop in semi-supervised settings due to its low generation accuracy, which is evident in Fig. 4b and Fig. 4e. Furthermore, CG-JEM, while being a closely related to the proposed method, performs roughly the same as vanilla CG and does not show significant performance improvement over other CGSGMs. We can also see in Fig. 4 and Table 2 that by incorporating the proposed self-calibration, all conditional metrics improved substantially. Notably, CG-SC-all (Ours)

Table 2: **Sample quality comparison of all methods with various percentages of labeled data.** **Bold**: best performance among all methods; underlined: best performance among CG-based methods. Den and Cov stands for Density and Coverage, respectively. CGSGMs demonstrate superior performance compared to CFSGMs in semi-supervised settings. Furthermore, the proposed CG-SC-labeled and CG-SC-all outperform other CGSGMs in semi-supervised settings.

(a) Results of semi-supervised settings on CIFAR-10 dataset

| Method | 5% labeled data | | | | 20% labeled data | | | |
|---|---|---|---|---|---|---|---|---|
| | intra-FID (↓) | FID (↓) | Den (↑) | Cov (↑) | intra-FID (↓) | FID (↓) | Den (↑) | Cov (↑) |
| CG-SC-labeled (Ours) | 24.93 | 2.84 | 1.083 | 0.816 | 16.62 | 2.75 | **1.097** | **0.823** |
| CG-SC-all (Ours) | **18.95** | 2.72 | **1.191** | **0.822** | **13.97** | 2.63 | 1.090 | 0.821 |
| CG | 31.17 | 2.61 | 1.047 | 0.815 | 24.94 | 3.09 | 1.004 | 0.806 |
| CG-JEM | 28.13 | 2.70 | 1.079 | 0.817 | 21.02 | 2.75 | 1.050 | 0.817 |
| CG-DLSM | 36.55 | **2.18** | 0.992 | 0.816 | 31.78 | **2.10** | 0.975 | 0.812 |
| CG-LS | 29.24 | 2.62 | 1.068 | 0.821 | 26.15 | 4.18 | 0.971 | 0.796 |
| CG-JR | 30.59 | 2.80 | 1.066 | 0.817 | 23.03 | 2.49 | 1.056 | 0.822 |
| Cond | 45.73 | 15.57 | 1.174 | 0.459 | 34.36 | 19.77 | 0.775 | 0.567 |
| CFG-labeled | 45.07 | 15.31 | 1.169 | 0.457 | 32.66 | 18.48 | 0.832 | 0.589 |
| CFG-all | 47.33 | 16.57 | 1.144 | 0.452 | 31.24 | 17.37 | 0.863 | 0.601 |

(b) Results of semi-supervised settings on CIFAR-100 dataset

| Method | 5% labeled data | | | | 20% labeled data | | | |
|---|---|---|---|---|---|---|---|---|
| | intra-FID (↓) | FID (↓) | Den (↑) | Cov (↑) | intra-FID (↓) | FID (↓) | Den (↑) | Cov (↑) |
| CG-SC-labeled (Ours) | 113.21 | 4.80 | 0.927 | 0.756 | 90.76 | 3.74 | 0.928 | 0.766 |
| CG-SC-all (Ours) | **101.75** | 4.31 | 0.968 | **0.770** | **80.42** | **3.60** | 0.941 | 0.775 |
| CG | 124.92 | 5.24 | 0.891 | 0.745 | 108.86 | 4.10 | 0.904 | 0.759 |
| CG-JEM | 127.47 | 6.01 | 0.831 | 0.723 | 113.16 | 5.08 | 0.864 | 0.754 |
| CG-DLSM | 134.11 | 4.46 | 0.862 | 0.749 | 126.12 | 7.24 | 0.802 | 0.718 |
| CG-LS | 118.52 | **4.18** | 0.933 | 0.762 | 103.39 | 3.70 | 0.963 | **0.778** |
| CG-JR | 119.78 | 4.64 | 0.926 | 0.759 | 105.91 | 3.92 | **0.941** | 0.771 |
| Cond | 129.82 | 10.58 | 1.210 | 0.404 | 111.73 | 29.45 | 0.622 | 0.451 |
| CFG-labeled | 133.03 | 11.25 | **1.220** | 0.385 | 117.09 | 32.68 | 0.611 | 0.424 |
| CFG-all | 133.18 | 10.68 | 1.210 | 0.380 | 113.38 | 30.84 | 0.612 | 0.437 |

consistently achieves the best conditional performance among all CGSGMs in semi-supervised settings. On average, CG-SC-all improves intra-FID by 11.60 and 25.81 over CG on CIFAR-10 and CIFAR-100, respectively. These results demonstrate that self-calibration enables CGSGMs to estimate the class-conditional distributions more accurately.

**Leverage unlabeled data for semi-supervised settings** As the proportion of unlabeled data increases, we expect this benefit of leveraging unlabeled data to become more significant. As our experimental results indicate in Fig. 4 and Table 2, the utilization of unlabeled data significantly improves performance. Notably, with only 5% of the data labeled on CIFAR-10, the performance of CG-SC-all evaluated with intra-FID and intra-Density are improved by 5.98 and 0.156 over CG-SC-labeled and 12.22 and 0.296 over the original CG. The results confirm that as the percentage of labeled data decreases, the benefit of utilizing unlabeled data becomes increasingly more profound.

## 5 Classifier-only score estimation

In this section, we present the results of classifier-only score estimation with 100% labeled data on the CIFAR-10 dataset to demonstrate the effectiveness of the proposed method further. Following JEM (Grathwohl et al., 2020), the proposed self-calibrated time-dependent classifiers can also be used as hybrid models

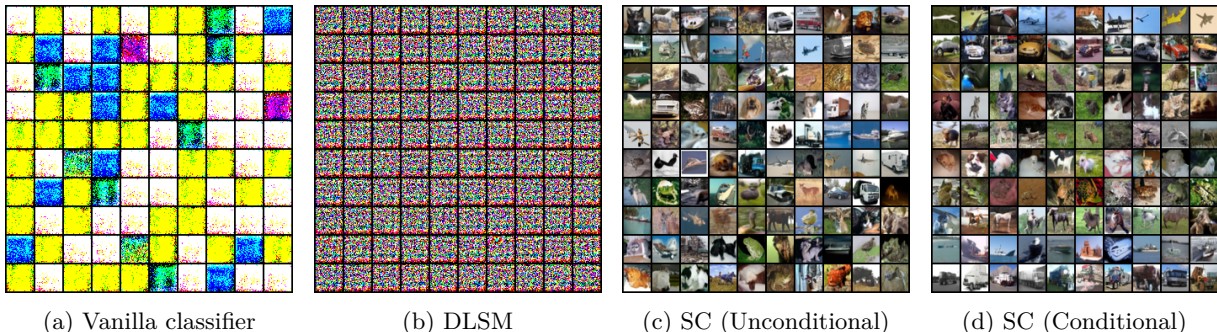

| (a) Vanilla classifier | (b) DLSM | (c) SC (Unconditional) | (d) SC (Conditional) |

Figure 5: **Generated images from classifier-only score estimation using different classifiers.** These results demonstrate that the incorporation of self-calibration loss enabled classifiers to estimate both conditional and unconditional scores.

that achieve both discriminative and generative modeling. By interpreting the time-dependent classifiers as SGMs, we can incorporate them in diffusion-based modeling for conditional and unconditional score estimations without requiring additional unconditional SGMs. We demonstrate in Section 5.1 and 5.2 the generative performance of such interpretation and how this interpretation could be used to improve adversarial robustness of classifiers. We view hybrid generative-discriminative modeling and improving adversarial robustness in the paradigm of diffusion models as interesting future research directions and hope this study can stimulate related explorations.

## 5.1 Classifier-only score-based generation

For unconditional generation, the unconditional score function estimated by classifiers, as shown in Eq. 7, is used for diffusion sampling; for conditional generation, both terms in the decomposed conditional score (Eq. 3) are estimated by classifiers. In other words, the time-dependent unconditional score $\nabla_{\boldsymbol{x}} \log p_t(\boldsymbol{x})$ can be written as

$$\nabla_{\boldsymbol{x}} \log p_t(\boldsymbol{x}) = \nabla_{\boldsymbol{x}} \log \Sigma_y \exp f(\boldsymbol{x}, y, t; \boldsymbol{\phi}) \tag{10}$$

where $f(\boldsymbol{x}, y, t; \boldsymbol{\phi})$ is the logits of the classifier. By adding the gradient of classifier $\nabla_{\boldsymbol{x}} \log p_t(y|\boldsymbol{x})$ to Eq. 10, we obtain the conditional score estimated by a classifier:

$$\begin{aligned}
\nabla_{\boldsymbol{x}} \log p_t(\boldsymbol{x}|y) &= \nabla_{\boldsymbol{x}} \log p_t(\boldsymbol{x}) + \nabla_{\boldsymbol{x}} \log p_t(y|\boldsymbol{x}) \\
&= \nabla_{\boldsymbol{x}} \log \Sigma_y \exp f(\boldsymbol{x}, y, t; \boldsymbol{\phi}) + \nabla_{\boldsymbol{x}} \log \frac{\exp\left(f(\boldsymbol{x}, y, t; \boldsymbol{\phi})\right)}{\sum_y \exp\left(f(\boldsymbol{x}, y, t; \boldsymbol{\phi})\right)} \\
&= \nabla_{\boldsymbol{x}} \log \Sigma_y \exp f(\boldsymbol{x}, y, t; \boldsymbol{\phi}) + \nabla_{\boldsymbol{x}} f(\boldsymbol{x}, y, t; \boldsymbol{\phi}) - \nabla_{\boldsymbol{x}} \log \Sigma_y \exp f(\boldsymbol{x}, y, t; \boldsymbol{\phi}) \\
&= \nabla_{\boldsymbol{x}} f(\boldsymbol{x}, y, t; \boldsymbol{\phi})
\end{aligned}$$

Here, the conditional score is essentially the gradient of the logits. Therefore, we sample from $\nabla_{\boldsymbol{x}} \text{LogSumExp}_y f(\boldsymbol{x}, y, t; \boldsymbol{\phi})$ for unconditional generation and $\nabla_{\boldsymbol{x}} f(\boldsymbol{x}, y, t; \boldsymbol{\phi})$ for conditional generation.

We present the generated images in Fig. 5. Without self-calibration, both the vanilla classifier (Fig. 5a) and DLSM (Fig. 5b) are unable to generate meaningful images when interpreted as conditional or unconditional SGMs. This inability demonstrates that without related regularization, the interpretation of classifiers as SGMs is not naturally learned through the time-dependent classification task. After regularizing the classifier the proposed self-calibration loss, Figures 5c and 5d show that not only does $\nabla_{\boldsymbol{x}} \text{LogSumExp}_y f(\boldsymbol{x}, y, t; \boldsymbol{\phi})$ become a more accurate estimator of unconditional score through direct training, $\nabla_{\boldsymbol{x}} f(\boldsymbol{x}, y, t; \boldsymbol{\phi})$ also becomes a better estimator of the conditional score. Besides generated images, we also include the quantitative measurements of conditional and unconditional classifier-only generation in Table 3. Although the proposed method demonstrates superior generative performance compared to other hybrid models, the performance is currently incomparable to state-of-the-art generative models.

Table 3: **Quantitative measurements of classifier-only generation.** Note that we compared the proposed method with results of hybrid models obtained by Grathwohl et al. (2020), and those results are obtained using a different model architecture. We calculated the classification accuracy of time-dependent classifiers by using the testing data with minimum perturbation, which corresponds to the minimum timestep. The proposed method achieves the best generation quality while having great discriminative performance.

| Method | FID ($\downarrow$) | IS ($\uparrow$) | Accuracy ($\uparrow$) | intra-FID ($\downarrow$) |
|---|---|---|---|---|
| Residual Flow (Chen et al., 2019) | 46.4 | 3.6 | 70.3% | |
| Glow (Kingma & Dhariwal, 2018) | 48.9 | 3.92 | 67.6% | |
| IGEBM (Du & Mordatch, 2020) | 37.9 | 8.3 | 49.1% | |
| JEM (Grathwohl et al., 2020) | 38.4 | 8.76 | **92.9%** | |
| SC (Unconditional) | 7.54 | **8.93** | 92.5% | |
| SC (Conditional) | **7.26** | **8.93** | 92.5% | 18.86 |

Table 4: **Original, adversarially attacked, and adversarial purified images.** Within four randomly selected images, all attacked images successfully led the classifiers to make incorrect predictions. Applying a single round of adversarial purification based on the proposed self-calibrated classifiers successfully removed the adversarial perturbations and led to correct predictions.

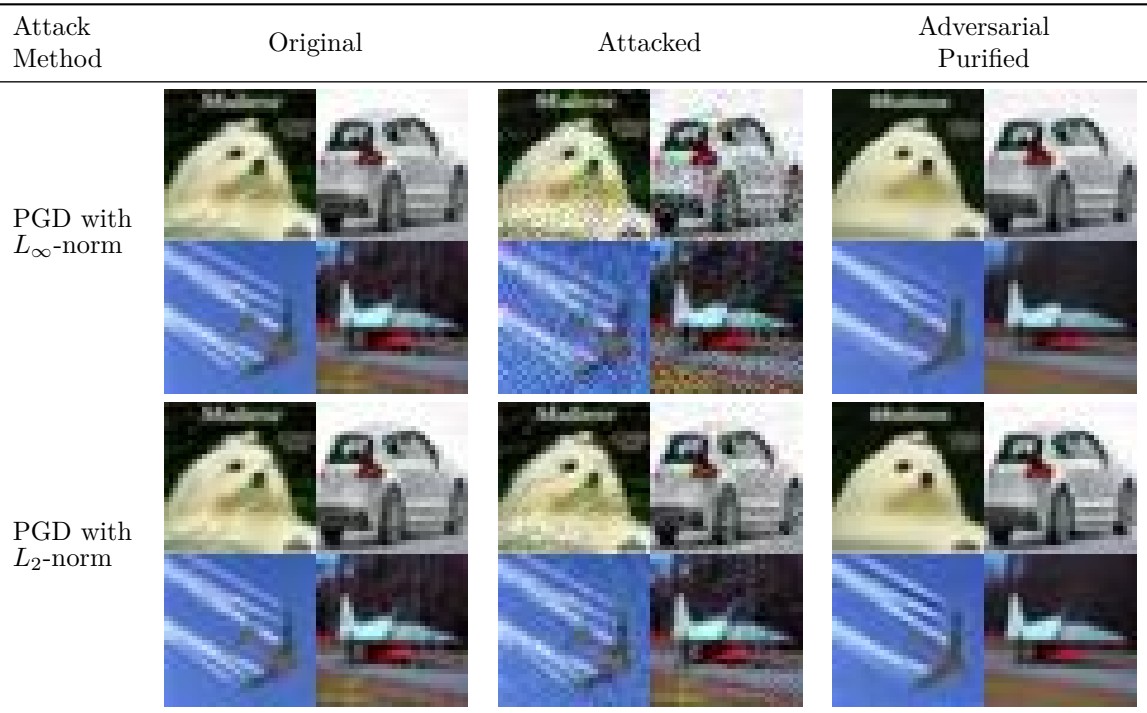

## 5.2 Improved adversarial robustness

In this section, we demonstrate how self-calibrated classifiers can be utilized to mitigate adversarial attacks. Grathwohl et al. (2020) showed that JEMs exhibit great resilience to such attacks. Additionally, JEM can perform adversarial purification by injecting noise and applying multiple steps of SGLD. We investigate whether similar results can be achieved within our framework.

To examine the adversarial robustness of classifiers, we used projected gradient descent (PGD) with $L_\infty$ and $L_2$ norms. Our experimental setup is identical to that described in JEM (Grathwohl et al., 2020), where 300 random test images are attacked using PGD with 20 random restarts and 40 iterations each. PGD

---

**Algorithm 3** Classification pipeline using classifier-only adversarial purification with the proposed self-calibrated classifier

---

**Input:** Clean/Perturbed sample $\boldsymbol{x}_0$, diffuse time $t$, #purification steps $s$, #purification rounds $r$
**Model:** Self-calibrated classifier $f(\boldsymbol{x}, y, t; \boldsymbol{\phi})$
Set $\boldsymbol{x}_0^0$ to be $\boldsymbol{x}_0$
**for** $i \leftarrow 1$ to $r$ **do**                                                    ▷ Apply $r$ rounds of purification
    Sample diffused sample $\boldsymbol{x}_t^i \sim p_t(\boldsymbol{x}_t | \boldsymbol{x}_0^{i-1})$
    Set $\boldsymbol{x}_{\text{prev}}$ to be $\boldsymbol{x}_t^i$
    **for** $j \leftarrow 1$ to $s$ **do**                           ▷ Obtain sample at timestep 0 by applying $s$ generation steps
        Sample $\boldsymbol{x}_{\text{cur}}$ from $\boldsymbol{x}_{\text{cur}} \sim p_{(1-j/s)t}\left(\boldsymbol{x}_{(1-j/s)t} | \boldsymbol{x}_{\text{prev}}\right)$ using score estimated with $\log \sum_y \exp f(\boldsymbol{x}, y, t; \boldsymbol{\phi})$
        Set $\boldsymbol{x}_{\text{prev}}$ to be $\boldsymbol{x}_{\text{cur}}$
    **end for**
    Set $\boldsymbol{x}_0^i$ to be $\boldsymbol{x}_{\text{cur}}$
**end for**
**Output:** Logits $f(\boldsymbol{x}_0^r, y, 0; \boldsymbol{\phi})$

---

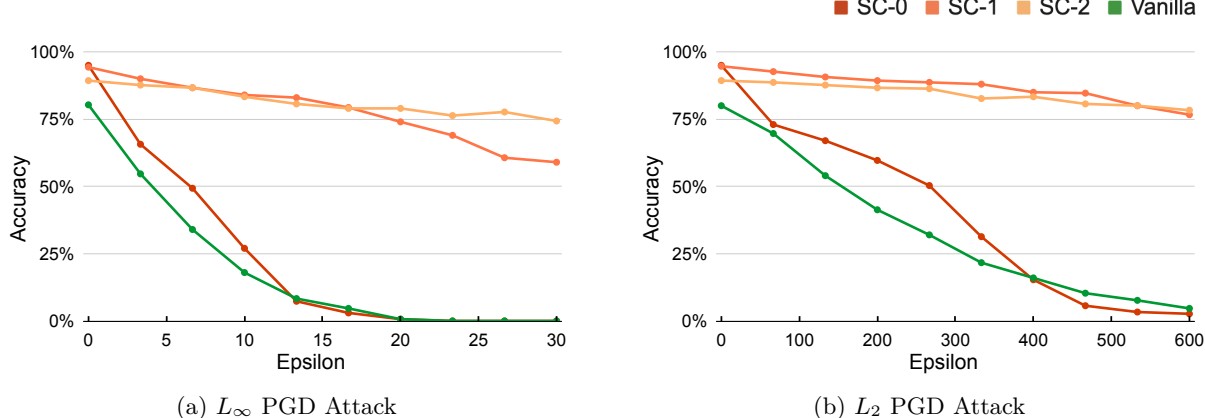

(a) $L_\infty$ PGD Attack
(b) $L_2$ PGD Attack

Figure 6: **Resilience of classifiers against adversarial attacks.** The figures contain classification accuracy after attacked by projected gradient descent (PGD) with $L_\infty$ and $L_2$ norms. The number after SC indicates the number of iterations of adversarial purification done to the attacked images.

attacks using $L_\infty$ and $L_2$ norms are applied with a maximum epsilon of 30 and 600, respectively. Inspired by JEM (Grathwohl et al., 2020) and DiffPure (Nie et al., 2022), we also implemented adversarial purification techniques that utilize classifier predictions. We treat the initial attacked images as $x_0$, perturb them to obtain $x_{\frac{T}{4}}$, and use reverse ODEs to eliminate adversarial perturbations. All classification accuracies are then evaluated at the minimum timestep $t = 0$. Algorithm 3 shows the classification pipeline with classifier-only adversarial purification. In our experiments, we drawed different $r$ from $\{0, 1, 2\}$ and set $(t, s) = (\frac{T}{4}, 1)$.

We show qualitative results of four randomly selected attacked images with maximum attack magnitudes in Table 4. The attacks injected considerable perturbation into the images and successfully tricked the classifiers into predicting incorrect labels. After performing adversarial purification, the perturbations are removed, making the classifiers able to classify purified images into the correct classes.

Fig. 6 presents the adversarial robustness of the classifiers before and after applying the proposed self-calibration technique. The number following SC indicates the rounds of adversarial purification applied; for example, SC-0 means no purification, and SC-2 means two rounds of purification. We observe that the vanilla time-dependent classifier, trained with cross-entropy loss on diffused data, already shows decent resilience against adversarial perturbation. Such resilience can be further enhanced by incorporating the proposed self-calibration (SC) loss, which improves the classifier's robustness through regularization.

In addition to this improvement, our method offers a unique benefit: classifier-only adversarial purification. Unlike previous studies that require an additional diffusion model to remove adversarial perturbations, the

proposed method enables us to utilize the attacked self-calibrated classifier itself to purify adversarial samples. This integration streamlines the defense pipeline by using the classifier for both adversarial purification and classification.

# 6 Conclusion

We tackle the overfitting issue for the classifier within CGSGMs from a novel perspective: self-calibration. The proposed self-calibration method leverages EBM interpretation like JEM to reveal that the classifier is internally an unconditional score estimator and design a loss with the DSM technique to calibrate the internal estimation. This self-calibration loss regularizes the classifier directly towards better scores without relying on an external score estimator. We demonstrate three immediate benefits of the proposed self-calibrating CGSGM approach. Using a standard synthetic dataset, we show that the scores computed using this approach are indeed closer to the ground-truth scores. Second, across all percentages of labeled data, the proposed approach outperforms the existing CGSGM. Lastly, our empirical study justifies that when compared to other conditional SGMs, the proposed approach consistently achieves the best intra-FID in the focused semi-supervised settings by seamlessly leveraging the power of unlabeled data, highlighting the rich potential of our approach.

# 7 Limitations and Future Directions

There are several future directions that we believe will further advance the effectiveness of the proposed method and the field of semi-supervised conditional generation:

- **More Comprehensive Theoretical Analysis**: While the proposed method is inspired by the sound and closely related frameworks of EBMs and SGMs, there is an opportunity to develop a formal theoretical analysis. Future studies could aim to theoretically justify how the self-calibration loss enhances classifier regularization and better aligns with the data distribution, thereby deepening our understanding of its underlying principles beyond empirical results.

- **Scaling to High-Resolution Images**: Our study is limited to 32x32 images due to computational constraints and the lack of well-trained unconditional SGMs for larger resolutions. Future studies could explore methodologies to overcome these limitations, substantiating the proposed method's applicability to more realistic settings.

- **Benchmarking of Semi-supervised Generative Models**: Given the rapid expansion of this research area, it is increasingly challenging to perform comprehensive experimental evaluations across all related methods and settings. This issue is particularly prohibitive for semi-supervised generative modeling, where a new percentage of labeled data requires re-running all methods. Future studies could aim to establish a systematic and comprehensive benchmarking framework that enables a unified and fair comparison of existing methods.

- **Most Promising Scenarios**: Our approach is particularly advantageous in semi-supervised settings where labeled data is scarce. The self-calibration loss effectively improves the accuracy of classifier gradients, thereby enhancing class-conditional generation. This makes the method well-suited for applications that require precise modeling of conditional distributions under limited supervision. Moreover, its demonstrated improvement in adversarial robustness positions it as a promising tool for security-critical domains, such as medical imaging or autonomous systems, where resistance to adversarial perturbations is essential.

- **Least Promising Scenarios**: Conversely, the current implementation may be less effective in scenarios where computational resources are more limited than human annotations. In these cases, the additional computational overhead introduced by the self-calibration process could limit its applicability. These limitations, however, represent opportunities for future research and optimization.

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

# A    Supplementary experimental results on CIFAR-10

## A.1    Additional semi-supervised learning settings

In Section 4, we discussed the generative performance using 5%, 20%, and 100% labeled data from CIFAR-10. In this section, we provide further results for scenarios where 40%, 60%, and 80% of the data is labeled. Besides the evaluation metrics used in the main paper, Inception Score (IS) (Salimans et al., 2016) is also provided. Note that results for CG-JEM with 40%, 60%, and 80% labeled data are not included due to limited computational resources.

Table 5: Sample quality comparison of all methods. **Bold**: best performance among all methods; underlined: best performance among CG-based methods.

(a) intra-FID, FID, and IS of the CIFAR-10 dataset

| Method | 5% labeled data | | | 20% labeled data | | | 100% labeled data | | |
|---|---|---|---|---|---|---|---|---|---|
| | intra-FID (↓) | FID (↓) | IS (↑) | intra-FID (↓) | FID (↓) | IS (↑) | intra-FID (↓) | Acc (↑) | IS (↑) |
| CG-SC-labeled (Ours) | 24.93 | 2.84 | 9.78 | 16.62 | 2.75 | 9.83 | 11.70 | 2.23 | 9.82 |
| CG-SC-all (Ours) | **18.95** | 2.72 | 9.95 | **13.97** | 2.63 | 9.94 | 11.70 | 2.23 | 9.82 |
| CG | 31.17 | 2.61 | 9.98 | 24.94 | 3.09 | 9.92 | 18.99 | 2.48 | 9.88 |
| CG-JEM | 28.13 | 2.70 | 9.92 | 21.02 | 2.75 | 10.10 | 23.39 | 2.16 | 9.83 |
| CG-DLSM | 36.55 | **2.18** | 9.76 | 31.78 | **2.10** | 9.91 | 21.59 | 2.36 | 9.92 |
| CG-LS | 29.24 | 2.62 | 9.92 | 26.15 | 4.18 | 9.98 | 18.10 | 2.15 | 9.98 |
| CG-JR | 30.59 | 2.80 | 9.84 | 23.03 | 2.49 | 10.04 | 17.24 | 2.17 | 9.89 |
| Cond | 45.73 | 15.57 | 9.87 | 34.36 | 19.77 | 8.82 | **10.29** | **2.13** | **10.06** |
| CFG-labeled | 45.07 | 15.31 | **10.20** | 32.66 | 18.48 | 8.93 | 10.58 | 2.28 | 10.05 |
| CFG-all | 47.33 | 16.57 | 9.89 | 31.24 | 17.37 | 9.15 | 10.58 | 2.28 | 10.05 |

| Method | 40% labeled data | | | 60% labeled data | | | 80% labeled data | | |
|---|---|---|---|---|---|---|---|---|---|
| | intra-FID (↓) | FID (↓) | IS (↑) | intra-FID (↓) | FID (↓) | IS (↑) | intra-FID (↓) | Acc (↑) | IS (↑) |
| CG-SC-labeled (Ours) | **12.08** | 2.78 | 10.00 | 11.65 | 2.37 | 9.91 | 11.86 | 2.24 | 9.78 |
| CG-SC-all (Ours) | 12.67 | 2.72 | 10.04 | 12.22 | 2.42 | 9.95 | 12.47 | 2.25 | 9.83 |
| CG | 18.31 | 2.42 | 9.95 | 16.94 | 2.35 | 10.03 | 20.15 | 3.30 | 9.76 |
| CG-DLSM | 29.33 | 2.35 | 9.85 | 23.52 | 2.15 | 9.83 | 21.76 | 2.30 | 9.96 |
| CG-LS | 17.89 | 2.32 | 9.95 | 17.72 | 2.27 | 9.91 | 22.30 | 2.40 | 9.84 |
| CG-JR | 18.63 | 2.43 | 10.01 | 19.05 | 2.25 | 10.06 | 18.36 | 2.15 | 9.90 |
| Cond | 13.65 | 4.36 | 9.94 | **10.93** | 2.55 | 10.00 | **10.61** | 2.37 | 10.03 |
| CFG-labeled | 13.93 | 4.59 | 9.84 | 11.28 | 2.73 | **10.12** | 10.75 | 2.48 | **10.09** |
| CFG-all | 13.43 | 4.30 | 9.98 | 11.38 | 2.83 | 10.05 | 10.94 | 2.50 | 10.03 |

(b) Density, Coverage, intra-Density, and intra-Coverage of the CIFAR-10 dataset

| Method | 5% labeled data | | | | 20% labeled data | | | | 100% labeled data | | | |
|---|---|---|---|---|---|---|---|---|---|---|---|---|
| | Den (↑) | Cov (↑) | intra-Den (↑) | intra-Cov (↑) | Den (↑) | Cov (↑) | intra-Den (↑) | intra-Cov (↑) | Den (↑) | Cov (↑) | intra-Den (↑) | intra-Cov (↑) |
| CG-SC-labeled (Ours) | 1.083 | 0.816 | 0.848 | 0.711 | **1.097** | **0.823** | 0.976 | 0.771 | 1.029 | 0.817 | 0.992 | 0.803 |
| CG-SC-all (Ours) | **1.191** | **0.822** | 1.040 | **0.757** | 1.090 | 0.821 | **1.020** | **0.789** | 1.029 | 0.817 | 0.992 | 0.803 |
| CG | 1.047 | 0.815 | 0.743 | 0.670 | 1.004 | 0.806 | 0.812 | 0.715 | 0.979 | 0.812 | 0.878 | 0.769 |
| CG-JEM | 1.079 | 0.817 | 0.794 | 0.691 | 1.050 | 0.817 | 0.871 | 0.739 | 0.980 | 0.813 | 0.781 | 0.723 |
| CG-DLSM | 0.992 | 0.816 | 0.630 | 0.634 | 0.975 | 0.812 | 0.688 | 0.680 | 0.943 | 0.803 | 0.779 | 0.734 |
| CG-LS | 1.068 | 0.821 | 0.777 | 0.685 | 0.971 | 0.796 | 0.789 | 0.708 | 1.005 | 0.818 | 0.861 | 0.767 |
| CG-JR | 1.066 | 0.817 | 0.758 | 0.677 | 1.056 | 0.822 | 0.841 | 0.727 | 1.007 | 0.815 | 0.881 | 0.766 |
| Cond | 1.174 | 0.459 | **1.180** | 0.445 | 0.775 | 0.567 | 0.736 | 0.552 | 1.050 | 0.831 | 1.040 | 0.827 |
| CFG-labeled | 1.169 | 0.457 | 1.159 | 0.442 | 0.832 | 0.589 | 0.777 | 0.569 | **1.101** | **0.838** | **1.092** | **0.831** |
| CFG-all | 1.144 | 0.452 | 1.129 | 0.440 | 0.863 | 0.601 | 0.829 | 0.585 | **1.101** | **0.838** | **1.092** | **0.831** |

| Method | 40% labeled data | | | | 60% labeled data | | | | 80% labeled data | | | |
|---|---|---|---|---|---|---|---|---|---|---|---|---|
| | Den (↑) | Cov (↑) | intra-Den (↑) | intra-Cov (↑) | Den (↑) | Cov (↑) | intra-Den (↑) | intra-Cov (↑) | Den (↑) | Cov (↑) | intra-Den (↑) | intra-Cov (↑) |
| CG-SC-labeled (Ours) | 1.080 | 0.820 | 1.040 | 0.810 | 1.000 | 0.810 | 0.962 | 0.796 | 1.030 | 0.820 | 0.992 | 0.803 |
| CG-SC-all (Ours) | 1.070 | 0.820 | 1.030 | 0.804 | 1.000 | 0.810 | 0.947 | 0.795 | 1.030 | 0.820 | 0.992 | 0.803 |
| CG | 1.030 | 0.820 | 0.909 | 0.769 | 0.920 | 0.800 | 0.817 | 0.750 | 0.980 | 0.810 | 0.878 | 0.769 |
| CG-DLSM | 0.980 | 0.810 | 0.782 | 0.732 | 0.940 | 0.810 | 0.781 | 0.737 | 0.940 | 0.800 | 0.779 | 0.734 |
| CG-LS | 1.030 | 0.820 | 0.894 | 0.768 | 0.970 | 0.810 | 0.788 | 0.737 | 1.010 | 0.820 | 0.861 | 0.767 |
| CG-JR | 1.020 | 0.820 | 0.858 | 0.752 | 0.990 | 0.810 | 0.853 | 0.760 | 1.010 | 0.820 | 0.881 | 0.766 |
| Cond | 1.050 | 0.820 | 1.040 | 0.816 | 1.050 | 0.830 | 1.040 | 0.821 | 1.050 | 0.830 | 1.040 | 0.827 |
| CFG-labeled | **1.100** | **0.830** | **1.089** | **0.822** | **1.120** | **0.840** | **1.111** | **0.834** | **1.100** | **0.840** | **1.092** | **0.831** |
| CFG-all | **1.100** | **0.830** | **1.089** | 0.820 | 1.110 | 0.830 | 1.103 | 0.830 | **1.100** | **0.840** | **1.092** | **0.831** |

Table 5 presents the results, further confirming the observations made in Section 4.2. The CFSGMs consistently exhibit high generation accuracy but suffer from significant performance drops as the labeled data percentage decreases. Conversely, the CGSGMs maintain stable performance across various settings. Furthermore, our proposed CG-SC consistently outperforms other CG-based methodologies in terms of intra-FID and generation accuracy.

## A.2 Evaluation of expected calibration error

Beyond the generative performance metrics, we present the Expected Calibration Error (ECE) to assess the calibration of classifiers regarding accurate probability estimation. ECE serves as a metric that evaluates the alignment of a classifier's confidence with its prediction accuracy. The classifier's confidence is defined as:

$$\mathrm{conf}(\boldsymbol{x}) = \max_{y} p(y|\boldsymbol{x}) = \max_{y} \frac{\exp(f(\boldsymbol{x}, y; \boldsymbol{\phi}))}{\sum_{y'} \exp(f(\boldsymbol{x}, y'; \boldsymbol{\phi}))},$$

where $f(\boldsymbol{x}, y; \boldsymbol{\phi})$ is the classifier's logits. We then divide the classifier's predictions based on confidence into several buckets. The average absolute difference between the confidence and prediction accuracy is calculated for each bucket. Then, given a labeled test set $D_t = \{(\boldsymbol{x}_m, y_m)\}_{m=1}^M$, ECE is defined as:

$$\text{ECE} = \sum_{i=1}^N \frac{|B_i|}{|D_t|} \cdot \left| \text{Acc}(B_i) - \frac{1}{|B_i|} \sum_{\boldsymbol{x} \in B_i} \text{conf}(\boldsymbol{x}) \right|,$$

where $N$ is the number of buckets, $B_i = \left\{ \boldsymbol{x} | \text{conf}(\boldsymbol{x}) \in [\frac{i-1}{N}, \frac{i}{N}) \right\}$, $\text{Acc}(B_i)$ is the averaged classification accuracy of $B_i$, and $\frac{1}{|B_i|} \sum_{\boldsymbol{x} \in B_i} \text{conf}(\boldsymbol{x})$ is the averaged confidence of $B_i$.

Table 6: Expected calibration error ($\downarrow$) of all methods with various percentages of labeled data

| Method | 5% | 20% | 40% | 60% | 80% | 100% |
|---|---|---|---|---|---|---|
| CG-SC-labeled (Ours) | 0.369 | 0.316 | **0.087** | **0.057** | **0.063** | **0.031** |
| CG-SC-all (Ours) | 0.210 | **0.243** | 0.102 | 0.109 | 0.111 | **0.031** |
| CG | 0.460 | 0.330 | 0.269 | 0.190 | 0.163 | 0.112 |
| CG-DLSM | 0.468 | 0.343 | 0.307 | 0.237 | 0.180 | 0.183 |
| CG-LS | **0.194** | 0.257 | 0.101 | 0.063 | 0.081 | 0.050 |
| CG-JR | 0.407 | 0.348 | 0.279 | 0.225 | 0.183 | 0.173 |

We follow the setup in Grathwohl et al. (2020), setting $N = 20$ for our calculations. The results are shown in Table 6, illustrating the ECE values for all CG-based methods across various percentages of labeled data. Our observations underscore that the self-calibration method consistently enhances classifier ECE in comparison to the vanilla CG and delivers the most superior ECE in most cases. This validates our claim that self-calibrated classifiers offer a more accurate probability estimation.

## B Run-time Benchmark for CG-SC (Ours), CG, CG-DLSM, and CG-JEM

We include a run-time benchmark of the four methods. The benchmark is done on CIFAR-10 with 100% labeled data over $200,000$ training steps using 4 V100 GPUs. We also provide these methods' additional run-time compared to vanilla CG.

Table 7: **Run-time benchmark for CG-SC (Ours), CG, CG-DLSM, and CG-JEM.** The experiments are performed on CIFAR-10 with 100% labeled data over $200,000$ training steps.

| Method | Run-time over 200k steps (s) | Additional Run-time (s) |
|---|---|---|
| CG-SC (Ours) | 129,935 | 80,998 |
| CG | 48,937 | - |
| CG-DLSM | 165,618 | 116,681 |
| CG-JEM | 1,213,258 | 1,164,321 |

From Table 7, we can see that the proposed CG-SC poses the least amount of additional computational cost among the three regularization methods.

## C More detailed introduction on score-based generative modeling through SDE

### C.1 Learning the score function

When learning the score function, the goal is to choose the best function from a family of functions $\{s(\boldsymbol{x}; \boldsymbol{\theta})\}_{\boldsymbol{\theta}}$, such as deep learning models parameterized by $\boldsymbol{\theta}$, to approximate the score function $\nabla_{\boldsymbol{x}} \log p(\boldsymbol{x})$ of interest. Learning is based on data $\{\boldsymbol{x}_n\}_{n=1}^N$ assumed to be sampled from $p(\boldsymbol{x})$. It has been shown that this can be

achieved by optimizing the in-sample version of the following score-matching loss over $\theta$:

$$\mathcal{L}_{\mathrm{SM}} = \mathbb{E}_{p(\boldsymbol{x})}\left[tr(\nabla_{\boldsymbol{x}} s(\boldsymbol{x};\boldsymbol{\theta})) + \frac{1}{2}\left\|s(\boldsymbol{x};\boldsymbol{\theta})\right\|_2^2\right],$$

where $tr(\cdot)$ denotes the trace of a matrix and $\nabla_{\boldsymbol{x}} s(\boldsymbol{x};\boldsymbol{\theta}) = \nabla_{\boldsymbol{x}}^2 \log p(x)$ is the Hessian matrix of log-likelihood $\log p(\boldsymbol{x})$. Calculating the score-matching loss requires $O(d)$ computation passes for $\boldsymbol{x} \in \mathbb{R}^d$, which makes the optimization process computationally prohibitive on high-dimensional data.

Several attempts (Kingma & Cun, 2010; Martens et al., 2012; Vincent, 2011; Song et al., 2019) have been made to address these computational challenges by approximating or transforming score matching into equivalent objectives. One current standard approach is called denoise score matching (DSM) (Vincent, 2011), which instead learns the score function of a noise-perturbed data distribution $q(\tilde{\boldsymbol{x}})$. DSM typically assumes that $q(\tilde{\boldsymbol{x}})$ comes from the original distribution $p(\boldsymbol{x})$ injected with a pre-specified noise $q(\tilde{\boldsymbol{x}}|\boldsymbol{x})$. It has been proved (Vincent, 2011) that the score function can be learned by minimizing the in-sample version of

$$\mathbb{E}_{q(\tilde{\boldsymbol{x}}|\boldsymbol{x})p(\boldsymbol{x})}\left[\frac{1}{2}\left\|s(\tilde{\boldsymbol{x}};\boldsymbol{\theta}) - \nabla_{\tilde{\boldsymbol{x}}}\log q(\tilde{\boldsymbol{x}}|\boldsymbol{x})\right\|_2^2\right],$$

where $\nabla_{\tilde{\boldsymbol{x}}}\log q(\tilde{\boldsymbol{x}}|\boldsymbol{x})$ is the score function of the noise distribution centered at $\boldsymbol{x}$. DSM is generally more efficient than the original score matching and is scalable to high-dimensional data as it replaces the heavy computation on the Hessian matrix with simple perturbations that can be efficiently computed from data.

### C.2 Generation from the score function by diffusion

Assume that we seek to sample from some unknown target distribution $p(\boldsymbol{x}) = p_0(\boldsymbol{x})$, and the distribution can be diffused to a known prior distribution $p_T(\boldsymbol{x})$ through a Markov chain that is described with a stochastic differential equation (SDE) (Song et al., 2021): $d\boldsymbol{x} = f(\boldsymbol{x},t)dt + g(t)d\boldsymbol{w}$, where the Markov chain is computed for $0 \le t < T$ using the drift function $f(\boldsymbol{x},t)$ that describes the overall movement and the dispersion function $g(t)$ that describes how the noise $\boldsymbol{w}$ from a standard Wiener process enters the system.

To sample from $p(\boldsymbol{x}) = p_0(\boldsymbol{x})$, Song et al. (2021) proposes to reverse the SDE from $p_T(\boldsymbol{x})$ to $p_0(\boldsymbol{x})$, which turns out to operate with another SDE (Eq. 1). Given the score function $s(\boldsymbol{x},t)$, the diffusion process in Eq. 1 can then be used to take any data sampled from the known $p_T(\boldsymbol{x})$ to a sample from the unknown $p(\boldsymbol{x}) = p_0(\boldsymbol{x})$.

The time-dependent score function $s(\boldsymbol{x},t;\boldsymbol{\theta})$ can be learned by minimizing a time-generalized (in-sample) version of the DSM loss because the diffusion process can be viewed as one particular way of injecting noise. The extended DSM loss is defined as

$$\mathcal{L}_{DSM}(\boldsymbol{\theta}) = \mathbb{E}_t\left[\lambda_t \mathbb{E}_{\boldsymbol{x}_t,\boldsymbol{x}_0}\left[\frac{1}{2}\left\|s(\boldsymbol{x}_t,t;\boldsymbol{\theta}) - s_t(\boldsymbol{x}_t|\boldsymbol{x}_0)\right\|_2^2\right]\right],$$

where $t$ is selected uniformly between 0 and $T$, $\boldsymbol{x}_t \sim p_t(\boldsymbol{x})$, $\boldsymbol{x}_0 \sim p_0(\boldsymbol{x})$, $s_t(\boldsymbol{x}_t|\boldsymbol{x}_0)$ denotes the score function of $p_t(\boldsymbol{x}_t|\boldsymbol{x}_0)$, and $\lambda_t$ is a weighting function that balances the loss of different timesteps.

## D Tuning the scaling factor for classifier guidance

This section includes the results when tuning the scaling factor $\lambda_{\mathrm{CG}}$ for classifier guidance with and without self-calibration under the fully-supervised setting.

Fig. 7 shows the result when tuning the scaling factor $\lambda_{\mathrm{CG}}$ for classifier guidance. When tuning $\lambda_{\mathrm{CG}}$ with and without self-calibration, self-calibration has little affect on unconditional performance. However, when evaluated with conditional metrics, the improvement after incorporating self-calibration becomes more significant. The improvement in intra-FID reaches 7.9 whereas generation accuracy improves by as much as 13%.

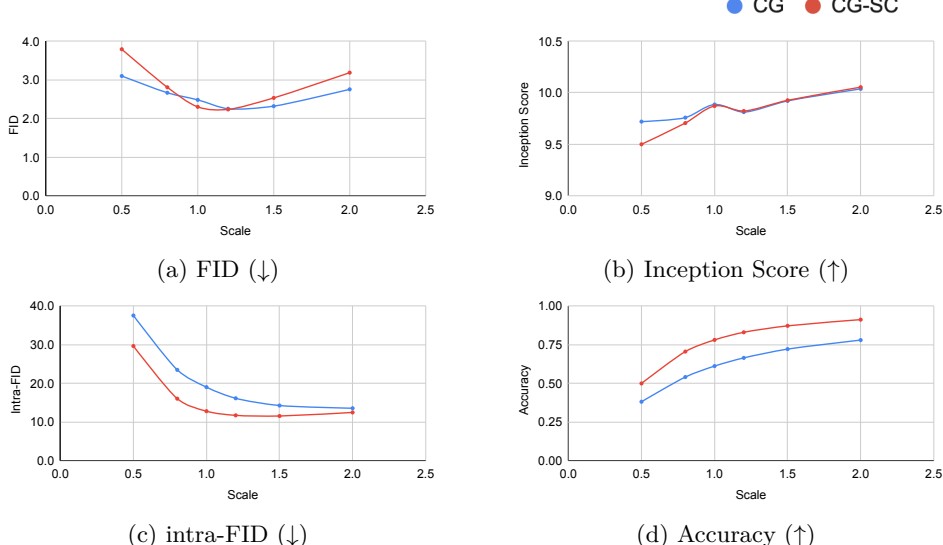

Figure 7: Results when tuning scaling factor $\lambda_{\text{CG}}$ for **CGSGM** (blue, without self-calibration) and **CGSGM-SC** (red, with self-calibration). (a) FID vs. $\lambda_{\text{CG}}$. (b) Inception score vs. $\lambda_{\text{CG}}$. (c) Intra-FID vs. $\lambda_{\text{CG}}$. (d) Generation accuracy vs. $\lambda_{\text{CG}}$. Unconditional metrics (FID and IS) differ little, but we observe a distinct performance gap when evaluated conditionally (intra-FID and accuracy).

## E  Test accuracies of time-dependent classifiers trained on CIFAR-10

Table 8: Test accuracies of various classifiers used by CG-based generation methods in Section 4.

| Method | 5% labeled data | 20% labeled data | 100% labeled data |
|---|---|---|---|
| CG-SC-labeled (Ours) | 0.430 | 0.675 | 0.925 |
| CG-SC-all (Ours) | 0.658 | 0.738 | 0.925 |
| CG | 0.420 | 0.571 | 0.808 |
| CG-JEM | 0.447 | 0.540 | 0.680 |
| CG-DLSM | 0.406 | 0.530 | 0.744 |
| CG-LS | 0.429 | 0.576 | 0.828 |
| CG-JR | 0.406 | 0.530 | 0.786 |

We showcase the test accuracies of different classifiers in Table 8. The test accuracy is evaluated at the minimum timestep $t = 0$ against unperturbed testing data. From the results, we can see that improvements in terms of classification accuracy is indeed observed after incorporating the proposed self-calibration. This further demonstrates the improvements over the baselines and confirms that the improvements in classifier guidance indeed originated from the improvements of time-dependent classifiers.

## F  Performance of the proposed CG-SC across various training steps

We include in this section the generative performance of the proposed CG-SC across different training steps on the CIFAR-10 dataset to showcase the stability of our method. We can see from Table F that the performance across different checkpoints of CG-SC is highly consistent. The result suggests that the proposed method is stable and reliable throughout training. Note that the result shown in Section 4 is sampled from the checkpoint at 300K training steps.

Table 9: Performance of different checkpoints of CG-SC trained on the CIFAR-10 dataset. Note that the result shown in Section 4 is sampled from the checkpoint at 300K training steps.

| Steps | FID (↓) | intra-FID (↓) | Density (↑) | Coverage (↑) | intra-Density (↑) | intra-Coverage (↑) | Acc (↑) |
|-------|---------|---------------|-------------|--------------|-------------------|--------------------|---------|
| 200K | 2.22 | 11.86 | 1.010 | 0.814 | 0.970 | 0.799 | 0.951 |
| 250K | 2.24 | 11.72 | 1.016 | 0.816 | 0.982 | 0.802 | 0.946 |
| 300K | 2.23 | 11.70 | 1.029 | 0.817 | 0.992 | 0.803 | 0.925 |
| 350K | 2.23 | 11.60 | 1.023 | 0.817 | 0.992 | 0.807 | 0.925 |
| 400K | 2.24 | 11.59 | 1.030 | 0.822 | 0.995 | 0.805 | 0.944 |
| 450K | 2.21 | 11.62 | 1.008 | 0.815 | 0.974 | 0.800 | 0.940 |
| 500K | 2.22 | 11.65 | 1.020 | 0.814 | 0.986 | 0.798 | 0.940 |
| 550K | 2.20 | 11.58 | 1.002 | 0.815 | 0.968 | 0.803 | 0.937 |
| 600K | 2.22 | 11.64 | 1.017 | 0.817 | 0.978 | 0.802 | 0.939 |
| 650K | 2.22 | 11.67 | 1.016 | 0.816 | 0.979 | 0.800 | 0.944 |

## G   Images generated by classifier guidance with and without self-calibration

This section includes images generated by classifier guidance with (first 6 images) and without (last 6 images) self-calibration after training on various percentages of labeled data. Each row corresponds to a class in the CIFAR-10 dataset. Generated images of all method can be found in the supplementary material.

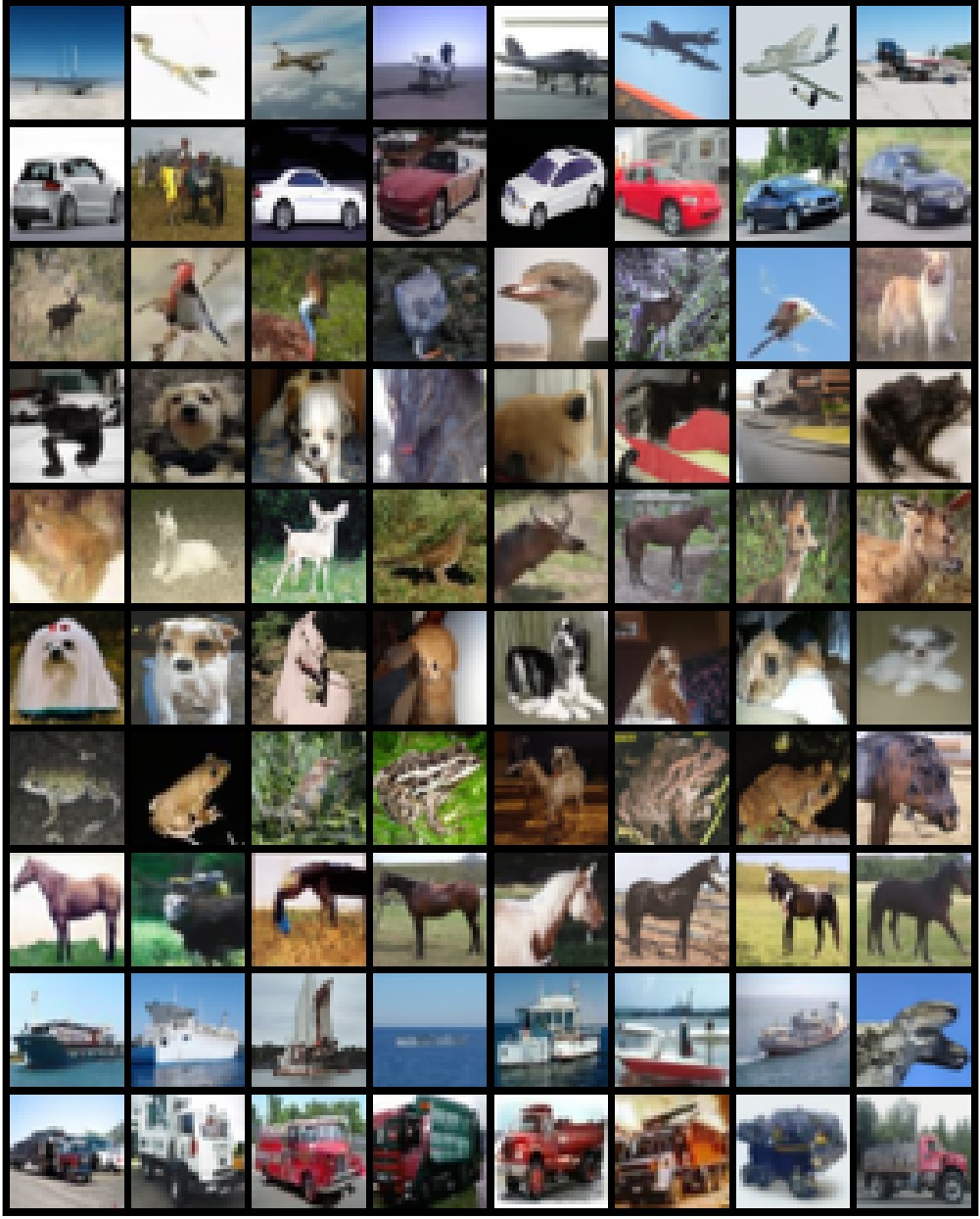

Figure 8: Randomly selected images of classifier guidance with self-calibration (5% labeled data)

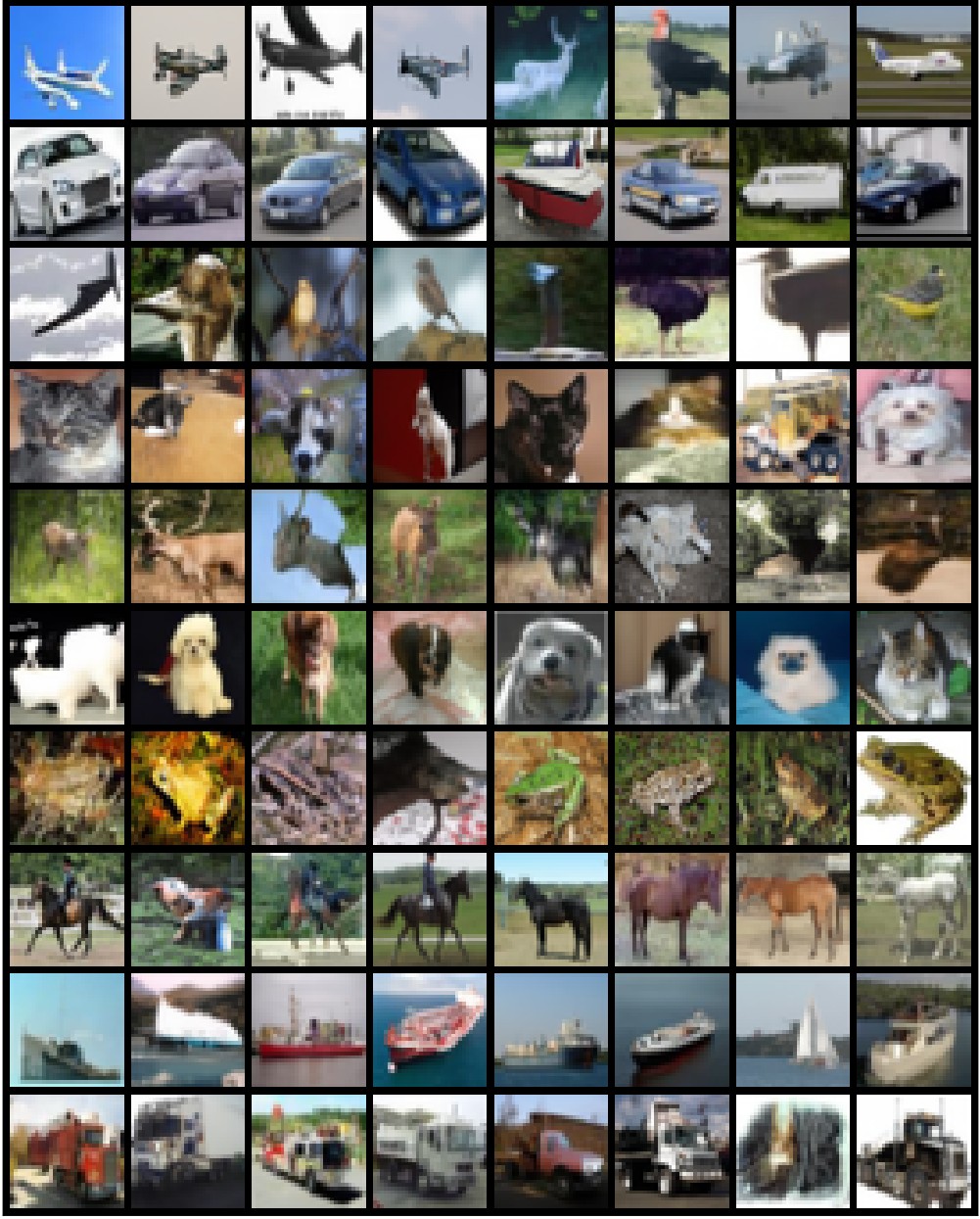

Figure 9: Randomly selected images of classifier guidance with self-calibration (20% labeled data)

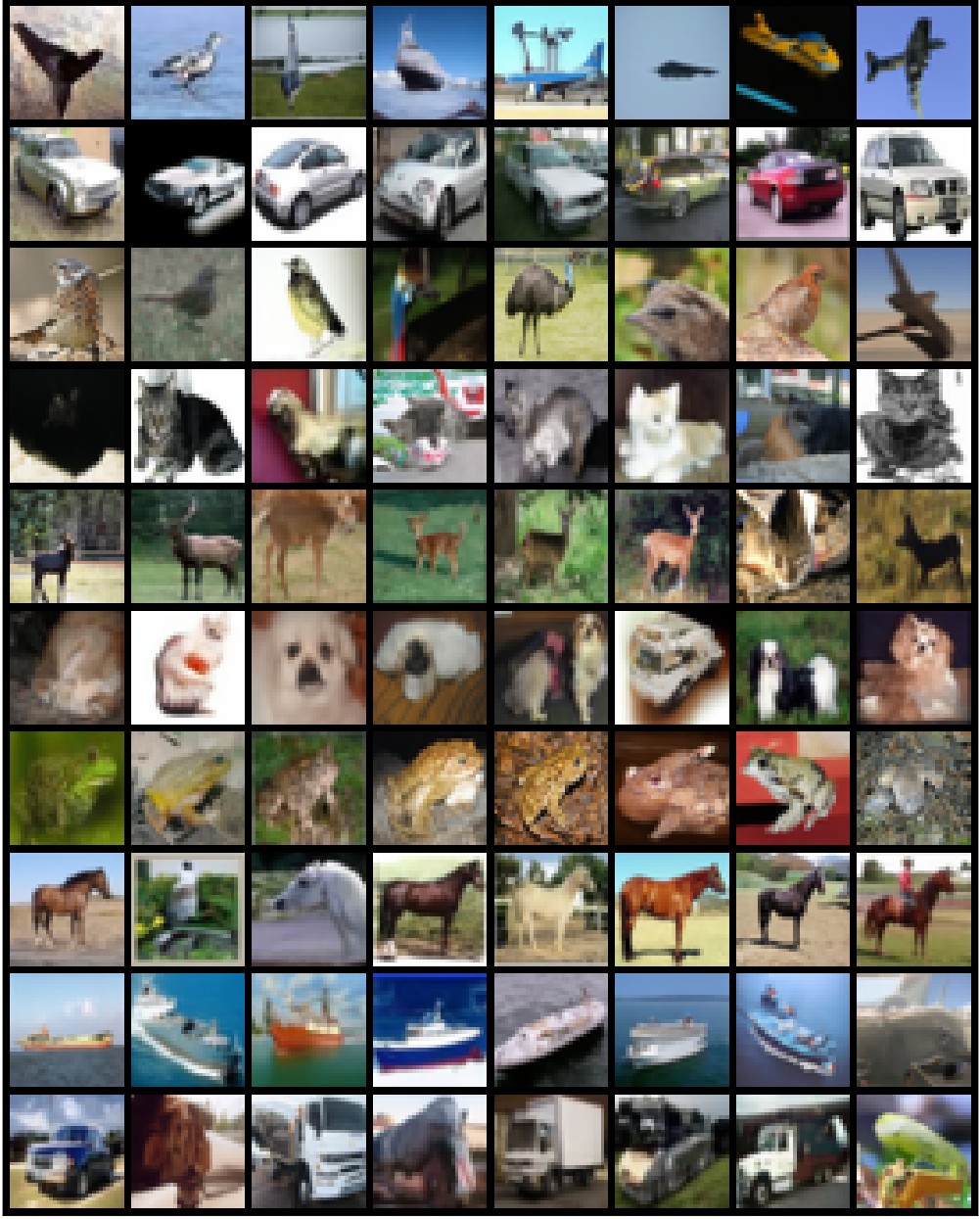

Figure 10: Randomly selected images of classifier guidance with self-calibration (40% labeled data)

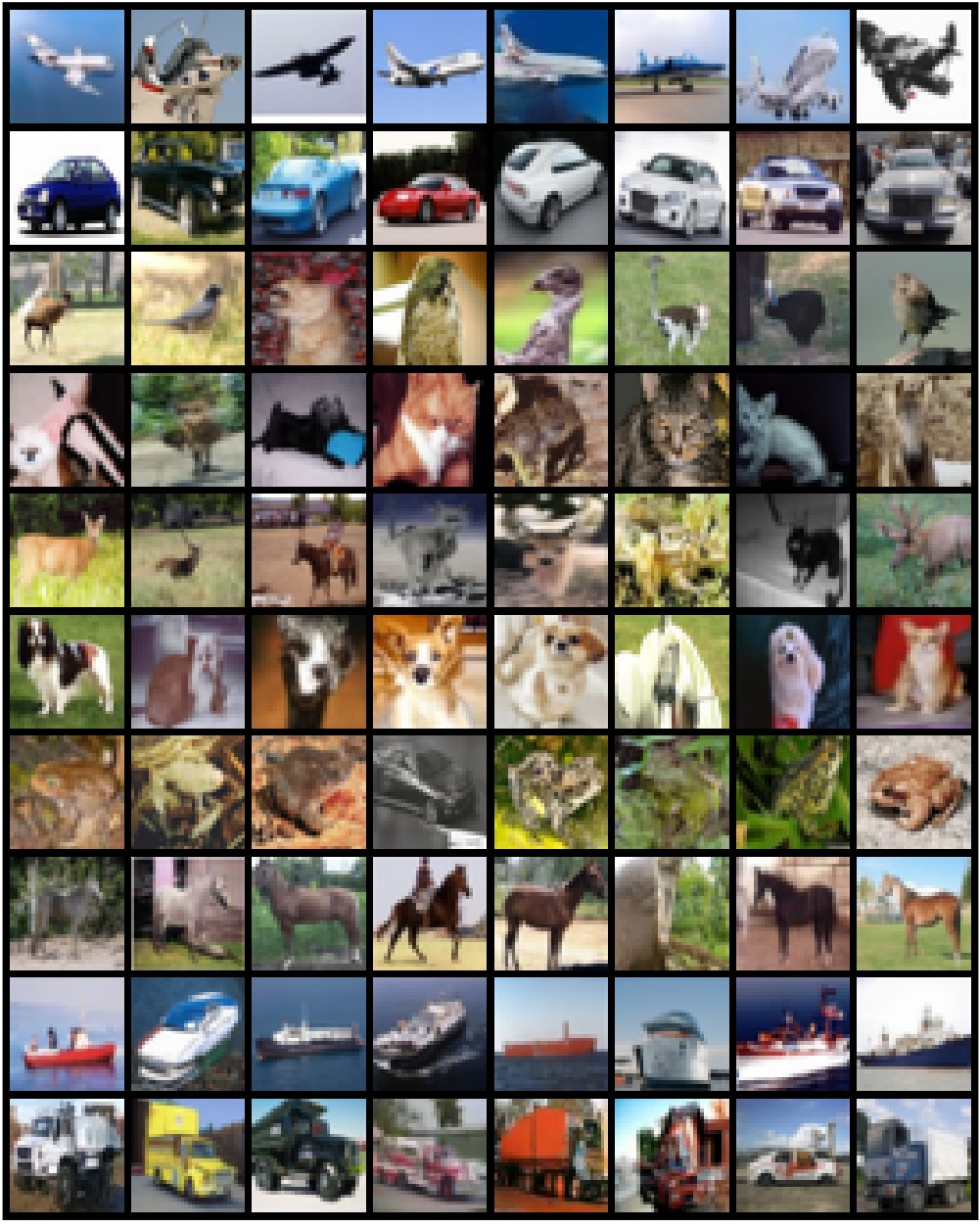

Figure 11: Randomly selected images of classifier guidance with self-calibration (60% labeled data)

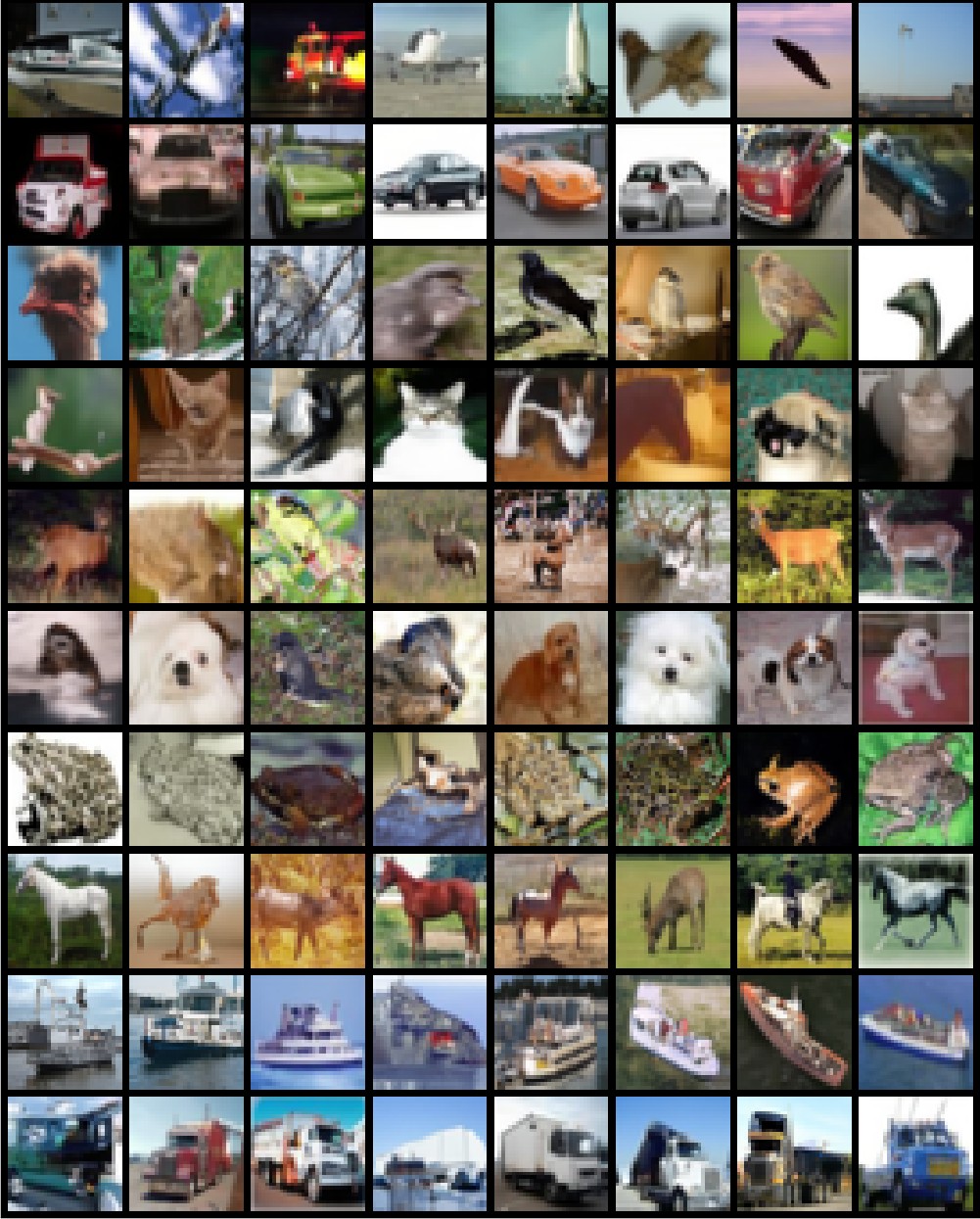

Figure 12: Randomly selected images of classifier guidance with self-calibration (80% labeled data)

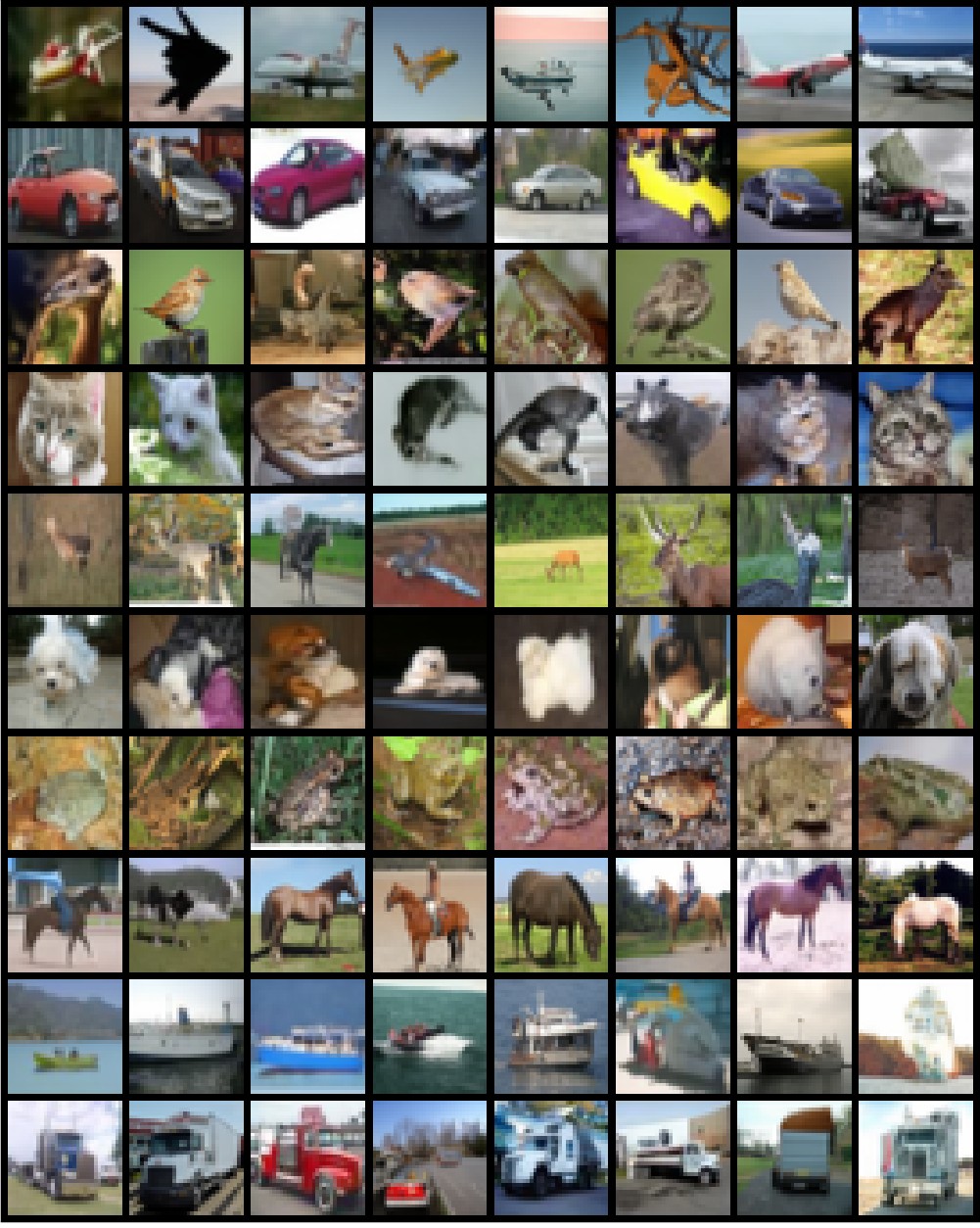

Figure 13: Randomly selected images of classifier guidance with self-calibration (100% labeled data)

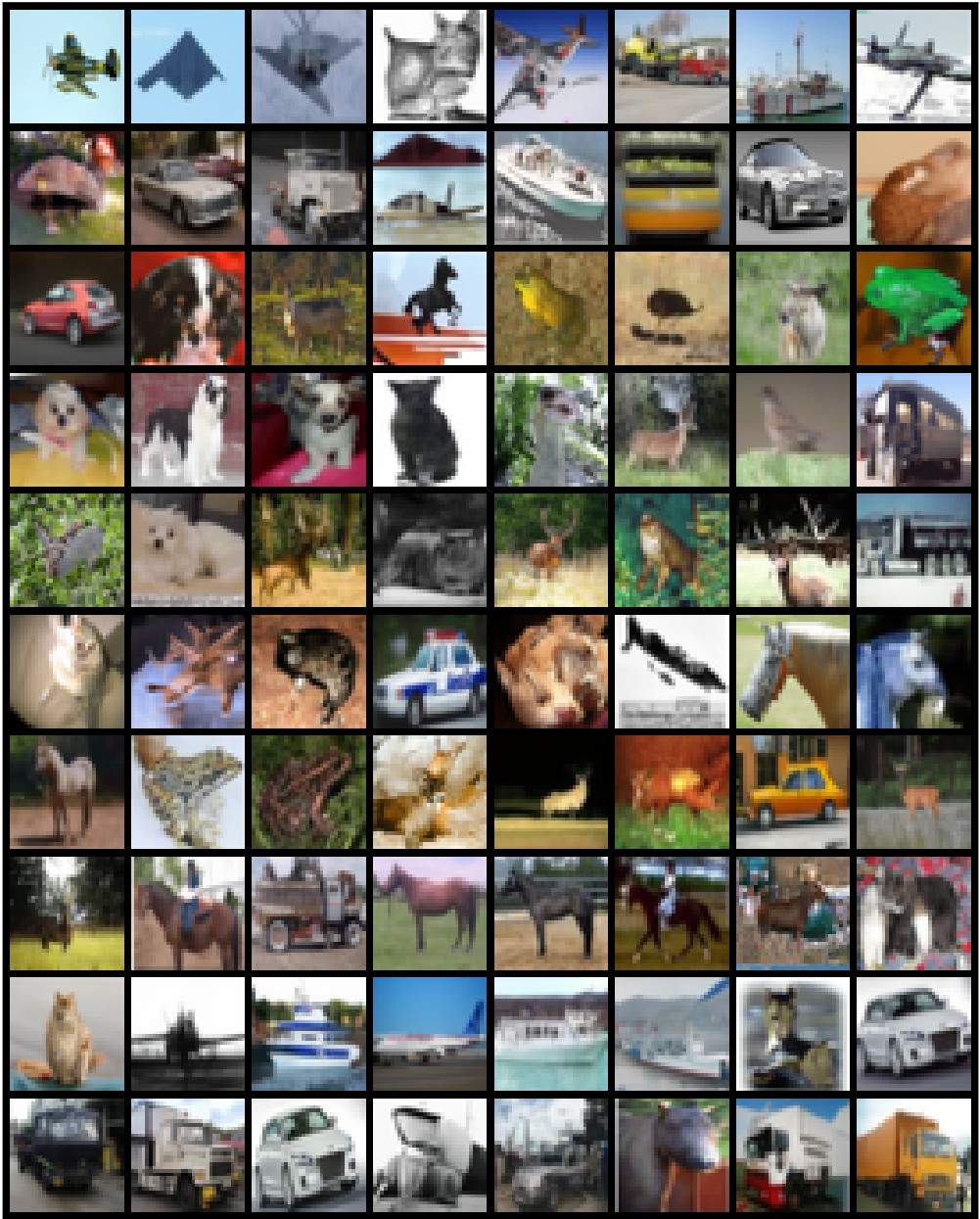

Figure 14: Randomly selected images of vanilla classifier guidance (5% labeled data)

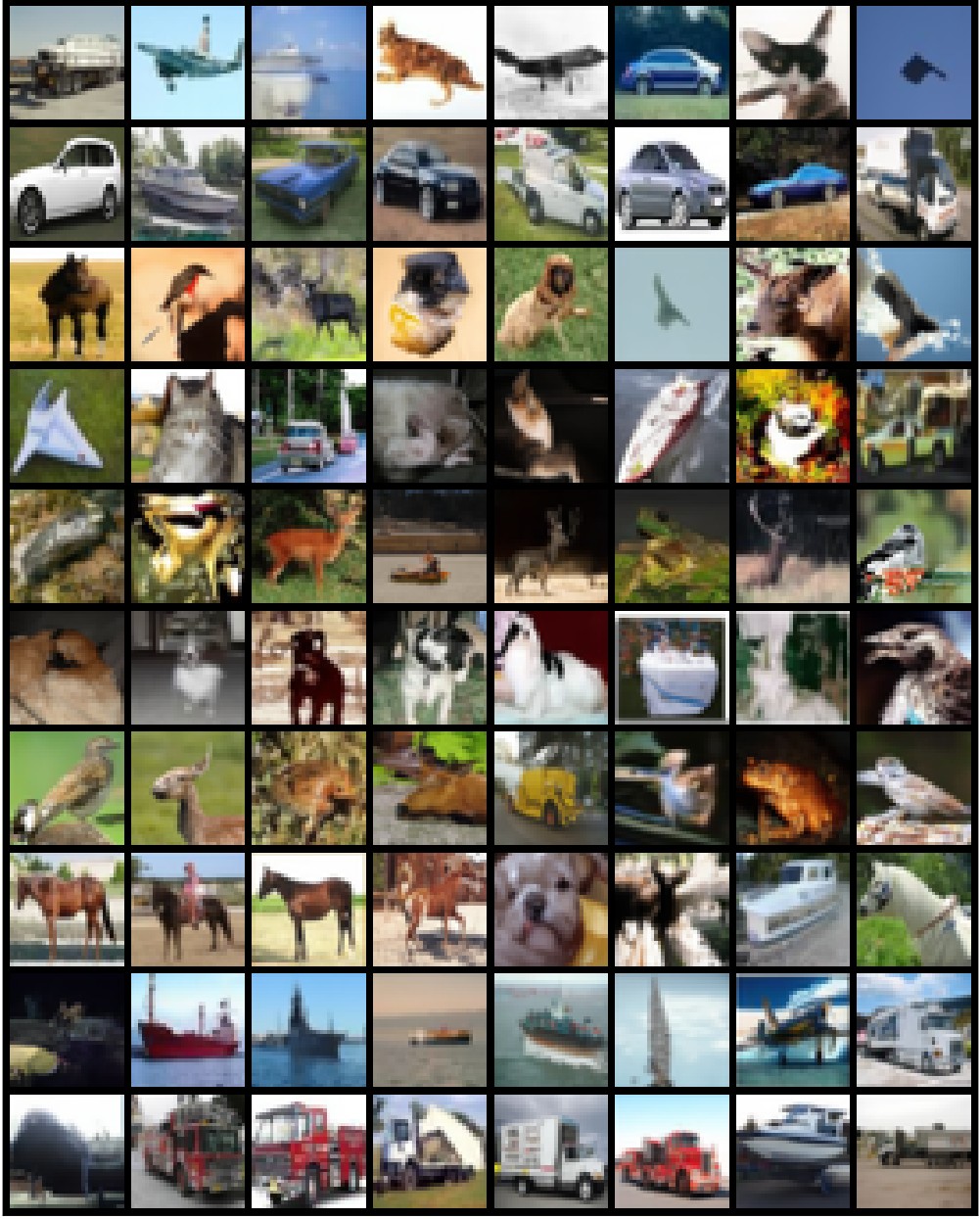

Figure 15: Randomly selected images of vanilla classifier guidance (20% labeled data)

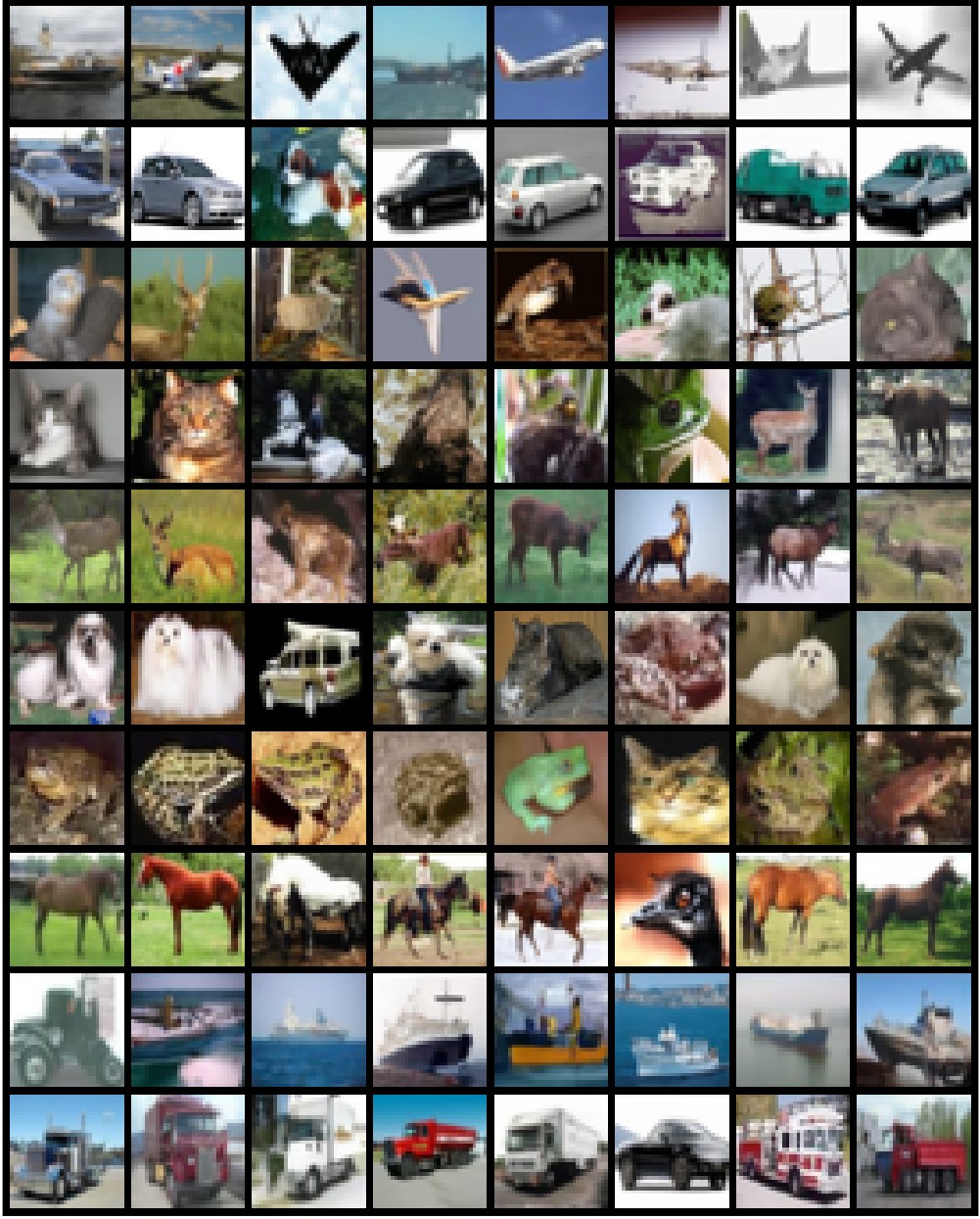

Figure 16: Randomly selected images of vanilla classifier guidance (40% labeled data)

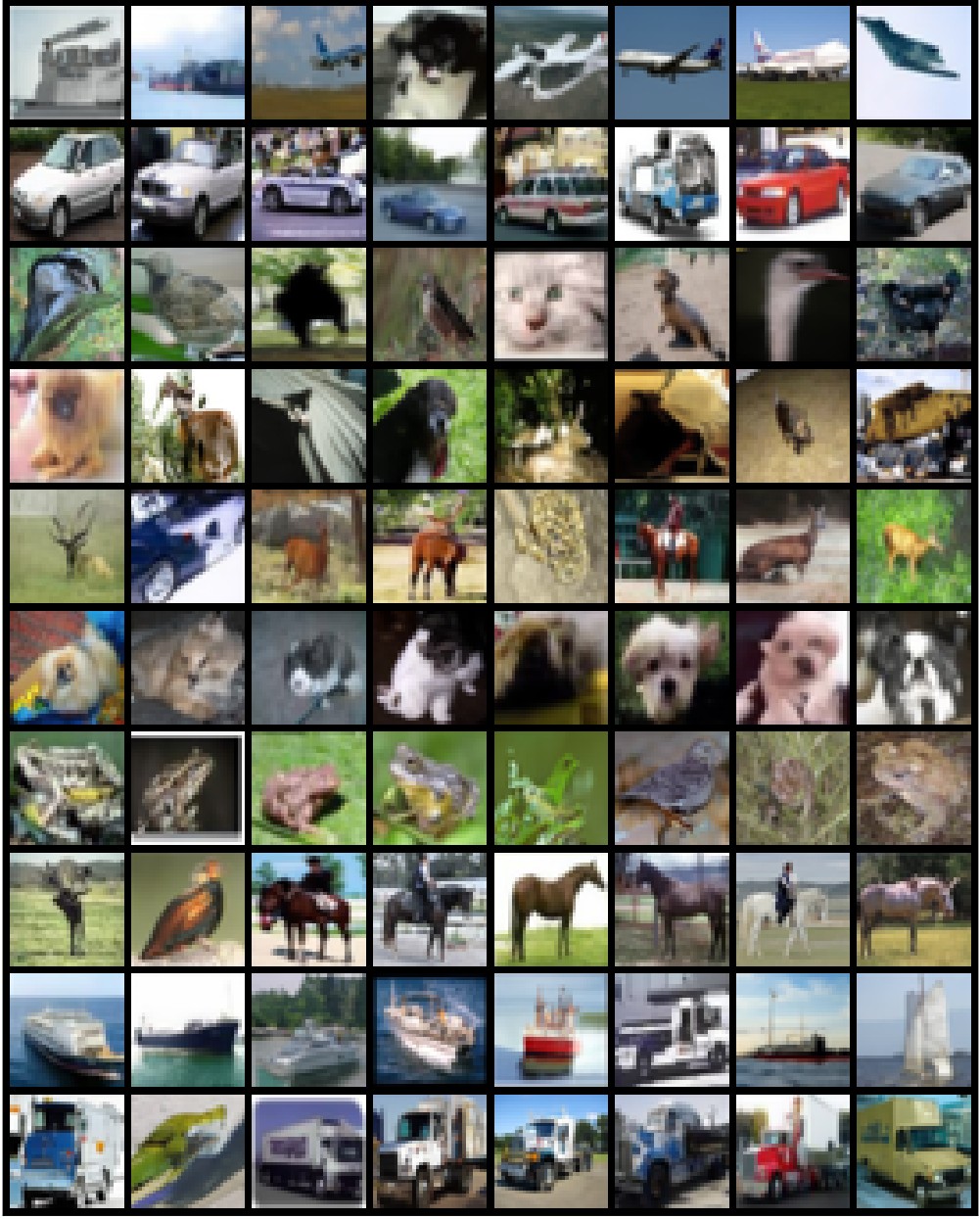

Figure 17: Randomly selected images of vanilla classifier guidance (60% labeled data)

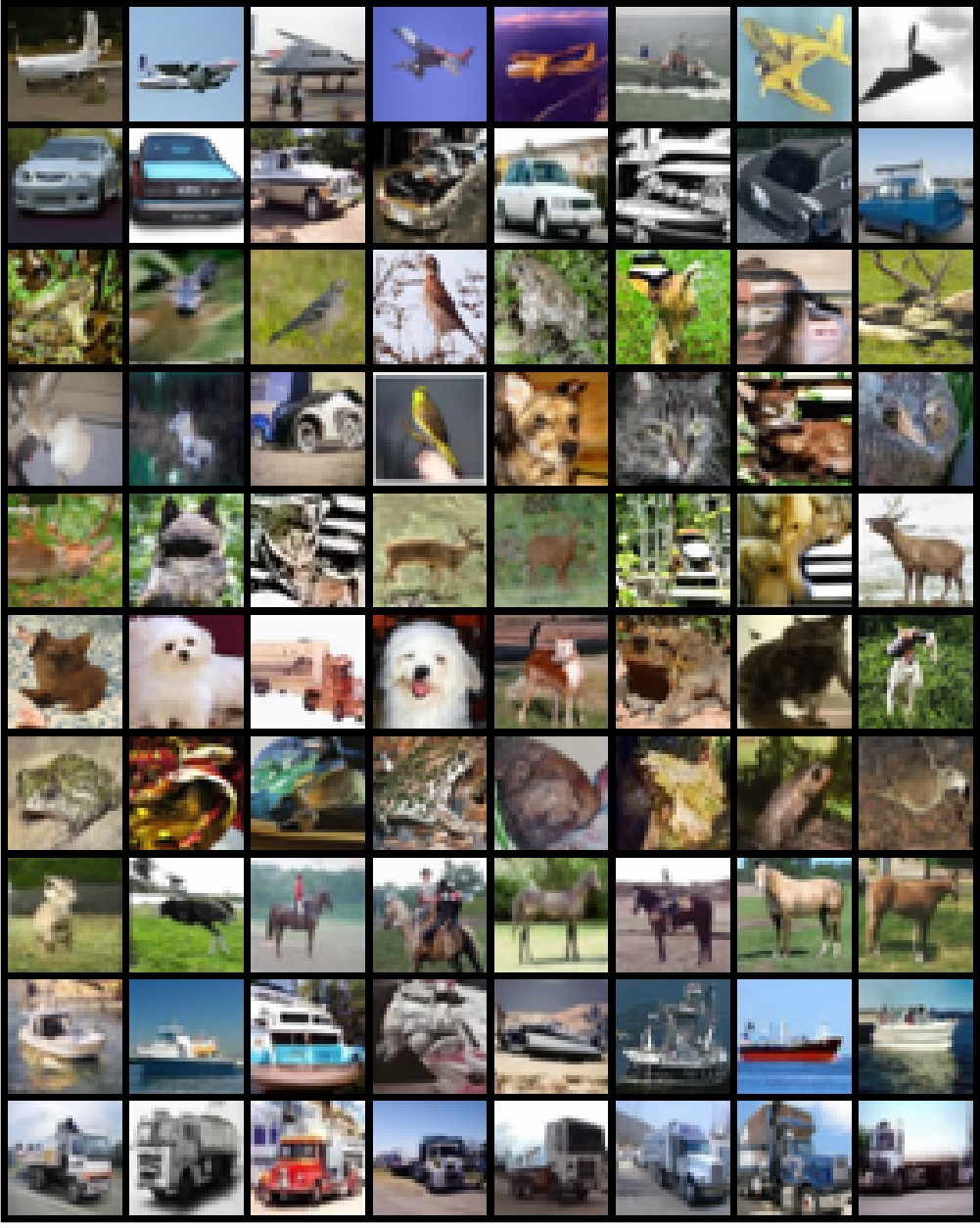

Figure 18: Randomly selected images of vanilla classifier guidance (80% labeled data)

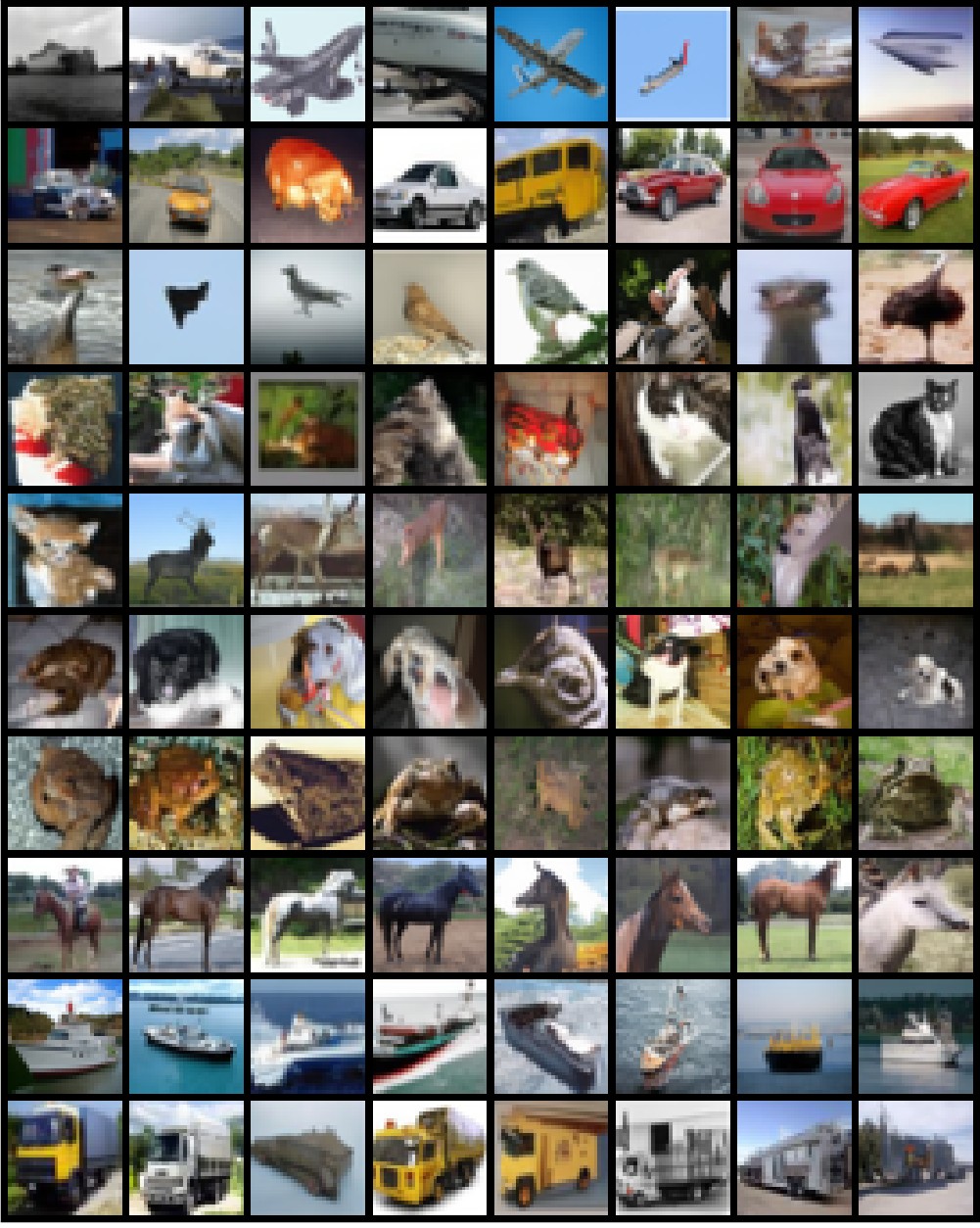

Figure 19: Randomly selected images of vanilla classifier guidance (100% labeled data)

