# OpenReview forum: "Semi-Supervised Classifier Guidance with Self-Calibration for Conditional Score-Based Generation"
_TMLR — Rejected by TMLR_

### Review · Reviewer_tVur · 2024-11-30

**Summary Of Contributions:**

This manuscript introduces a semi-supervised approach for conditional score-based generative models (diffusion models). The key idea is integrating unlabeled data to train classifier guidance schemes under limited label scenarios. To be specific, it proposes to utilize the Joint Energy-based Model (JEM) allows training on labeled and unlabeled data by replacing the classifier. Then, the authors introduce self-calibration loss to match scores between JEM and the diffusion model during training. Throughout extensive experiments on the CIFAR scales, the authors demonstrate its effectiveness compared to existing classifier-guidance and classifier-free based approaches.

**Audience:**

Yes

**Claims And Evidence:**

Yes

**Requested Changes:**

See the above

**Strengths And Weaknesses:**

**Strengths**
- The paper is well-structured and written in a way that is easy to understand.
- The idea of utilizing JEM is interesting.
- It has shown significant improvements in intra-FID style metrics on CIFAR10 and CIFAR100.

**Weaknesses**
- The primary weakness of this paper is the lack of comparison with the most relevant methodologies. Semi-supervised learning approaches for conditional generation have been extensively studied in VAE [1], GAN [2,3], and diffusion models [4]. However, this paper does not incorporate or reflect these baselines at all.
- There are various methods to integrate unlabeled data into conditional SGMs beyond the proposed approach. For instance, the above-mentioned paper [4] employs a two-step process: first training a semi-supervised classifier, and then using the pseudo-labels to train a conditional generative model. This paper, however, lacks a sufficient discussion on how to train a semi-supervised classifier. Furthermore, it fails to adequately justify why the proposed JEM-based methodology (with SC loss) is superior to alternative approaches.
- Does JEM guidance outperform classifier guidance? How can this be verified? Even when using JEM, why is JEM Score guidance superior to JEM classifier guidance? These critical points are not discussed in Table 3. Additionally, JEM classifier guidance should be compared with semi-supervised baselines, as shown in Tables 1 and 2, and this comparison must be thoroughly explained.
- In Table 2, it is challenging to confirm the proposed methodology's superiority in terms of FID. For example, it performs significantly worse than the CG DLSM baseline and is generally inferior. Given the importance of FID, the interpretation and explanation of these results are insufficient.
- In the analysis related to adversarial robustness, it remains unclear whether the observed tendency stems from the proposed methodology or represents a general characteristic of score-based generative models. This ambiguity should be clarified.

[1]Semi-supervised learning with deep generative models

[2] Triple generative adversarial nets

[3] High-fidelity image generation with fewer labels

[4] Diffusion Models and Semi-Supervised Learners Benefit Mutually with Few Labels

---

> ### Author Response · Authors · 2025-02-13
> **Response to Reviewer tVur by Authors Part 1**
>
> > The primary weakness of this paper is the lack of comparison with the most relevant methodologies. Semi-supervised learning approaches for conditional generation have been extensively studied in VAE [1], GAN [2,3], and diffusion models [4]. However, this paper does not incorporate or reflect these baselines at all.
>
> Thank you for pointing this out. In response to this valuable suggestion, we plan to include a thorough literature comparison to those methodologies in the revision in the coming week, highlighting the strength and weakness of each methodology. Given the two week limit of responding, we will unfortunately not be able to carry out more experimental comparisons, and we will include this limitation in our discussions in the revision. Nevertheless, we believe this does not change the fact that our detailed empirical results and ablation studies provide accurate, convincing, and clear evidence in support of our claim that SC loss is promising within CGSGMs.
>
> > This paper, however, lacks a sufficient discussion on how to train a semi-supervised classifier.
>
> Thank you for pointing out the need to clarify this. In the revision in the coming week, we plan to include a more detailed introduction to the training process for time-dependent classifiers and the proposed method under semi-supervised settings. Furthermore, we plan to include a detailed comparison between the training procedures of SC (ours), DLSM, and [1].
>
> [1] Diffusion Models and Semi-Supervised Learners Benefit Mutually with Few Labels
>
> > Does JEM guidance outperform classifier guidance? How can this be verified? Even when using JEM, why is JEM Score guidance superior to JEM classifier guidance? These critical points are not discussed in Table 3.
>
> Thanks for the question. In our experiments, JEM guidance (CG-JEM) performs around the same as vanilla classifier guidance. This is because training with MCMC sampling is highly unstable, which is significantly worsened when coupled with other training objectives (such as cross-entropy loss in our case). Such instability makes it hard for the training losses to converge to a low and steady stage. Although increasing the MCMC sampling step would improve its stability, doing so would make the computational cost prohibitive during training. In contrast, the proposed method circumvents the need for MCMC sampling during training, thus improving both stability and efficiency. We will update our manuscript to include related discussion in the revision.
>
> > JEM classifier guidance should be compared with semi-supervised baselines, as shown in Tables 1 and 2, and this comparison must be thoroughly explained.
>
> Thanks for the suggestion. We plan to include the suggested comparison in the revision.
>
> > In Table 2, it is challenging to confirm the proposed methodology's superiority in terms of FID. For example, it performs significantly worse than the CG DLSM baseline and is generally inferior. Given the importance of FID, the interpretation and explanation of these results are insufficient.
>
> Thank you for checking this. Yes, factually we agree that the proposed methodology does not perform the best in terms of the global FID. However, please note that the global FID is not the most expressive metric in the conditional generation setting, which generates examples of multiple modes (per class) instead of a single (mean, co-variance) mode that FID evaluates. Under the conditional generation setting, intra-FID, which is measured per class, should be much more meaningful than the global FID. Our proposed methodology demonstrates improved intra-FID compared to all other CGSGMs, showcasing its superior effectiveness in modeling class-conditional distributions. Significant improvements in the intra-Density metric also demonstrate our method's ability to generate samples corresponding to correct classes while having limited supervision; in contrast, other CGSGMs tend to perform significantly worse in terms of intra-Density, which indicates that generated samples tend to be out-of-distribution for class-conditional distributions of the training data. Those being said, the mild change in terms of the global FID should not be worrisome in our humble opinion. In the revision, we will include detailed discussions on the interpretation of these evaluation metrics for clarification.

---

> ### Author Response · Authors · 2025-02-13
> **Response to Reviewer tVur by Authors Part 2**
>
> > In the analysis related to adversarial robustness, it remains unclear whether the observed tendency stems from the proposed methodology or represents a general characteristic of score-based generative models. This ambiguity should be clarified.
>
> Thank you for pointing out the ambiguity. The benefit of our proposed method in defending against adversarial attacks can mainly be categorized into the following two elements:
> - **Improved adversarial robustness against adversarial attacks**: In Fig. 6, vanilla time-dependent classifiers demonstrate decent resilience against adversarial attacks because of the outstanding robustness obtained through training on diffused data. Furthermore, we can further improve such resilience and robustness by incorporating the proposed SC loss, which is evident in Fig. 6. Therefore, we conclude that time-dependent classifiers naturally possess decent adversarial robustness, which can be further improved by the proposed method.
> - **Classifier-only adversarial purification**: Previous studies have demonstrated the potential of using diffusion models to remove adversarial perturbations. However, to the best of our knowledge, all of them require using a separate diffusion model. In contrast, our methodology allows for using the attacked classifier itself to remove adversarial purifications, circumventing the need for an extra diffusion model in the defense pipeline.
>
> In the revision, we plan to include a more detailed discussion for clarification.

---

### Review · Reviewer_wDV4 · 2025-01-16

**Summary Of Contributions:**

The manuscript proposes an alternative method for classifier guidance for score-based generative models such as the popular diffusion models. The authors train a classifier on the time-dependent diffusion samples and interpret the classifier as a time-dependent energy-based model. By combining the regular cross-entropy loss with a second time-depedendent "self-calibration" score-matching loss (matching the gradient of the sum of the classifier logits to the true score), the classifier is regularized to have better gradients for the class-conditional generation.
The results show this method performing well in terms of deviation from the correct gradient, FID, and other metrics like coverage on CIFAR10 and CIFAR100.
Furthermore, they show that the classifier trained with their self-calibration loss is more adversarially robust against L-inf or L2-norm-bounded Projected Gradient descent attacks than a regaularly trained classifier.

**Audience:**

Yes

**Claims And Evidence:**

Yes

**Requested Changes:**

Maybe add a bit about training time and inference time as well, also compared to other approaches. Or was it somewhere and I missed it?

And as written above maybe add which scenarios you believe your method is most and least promising to use.

Also, did I understand everything correctly as per what  I have written in my review? Classifier gets time-dependently noised/diffused samples, is trained on them with cross-entropy and score-matching loss (based on gradient of sum of logits)? I think Figure 2 and caption explain this fairly well, but saw it quite late, maybe a similar text could be integrated somewhere in the main text as well, or figure 2 maybe shifted more towards section 3, so it can be seen alongside more easily?

**Strengths And Weaknesses:**

* The manuscript presents a clearly motivated idea with a clear mathematical derivation.
* The connection to existing work is very explicit and nicely explained.
* I find the figures are well-done and the writing relatively good to read.
* To the best of my knowledge, the idea is novel and conceptually sound.
* Experimental evaluation nicely showcases both settings where method outperforms and others and also where not
* Extension to adversarial robustness seems convincing to me, would of course benefit from further comparison to other adversarial robustness methods, but that might make it go out of scope


The authors could explain a bit more overall, in which scenarios they feel their method is most and least promising to try/use.

---

> ### Author Response · Authors · 2025-02-13
> **Response to Reviewer wDV4 by Authors**
>
> > Extension to adversarial robustness seems convincing to me, would of course benefit from further comparison to other adversarial robustness methods, but that might make it go out of scope
>
> Thank you. Given that we use a different architecture than the current adversarial training works and the two-week time constraint, we plan to implement only the most representative methods on our model architecture as a proof of concept. In particular, we plan to run AutoAttack against the proposed method. We expect to include the results in Section 5 in the coming week.
>
> > Maybe add a bit about training time and inference time as well, also compared to other approaches. Or was it somewhere and I missed it?
>
> Thank you for the suggestion. We totally agree with this, and before the coming week, we will include discussions on the computational efficiency of CG, CG-DLSM, and the proposed method in the appendix of the revision.
>
> > And as written above maybe add which scenarios you believe your method is most and least promising to use.
>
> We appreciate your suggestion and provide the following discussions regarding the scenarios in which our method is most and least promising. The discussions will be included in the revision.
>
> - **Most Promising Scenarios**:
>   Our approach is particularly advantageous in semi-supervised settings where labeled data is scarce. The self-calibration loss effectively improves the accuracy of classifier gradients, thereby enhancing class-conditional generation. This makes the method well-suited for applications that require precise modeling of conditional distributions under limited supervision. Moreover, its demonstrated improvement in adversarial robustness positions it as a promising tool for security-critical domains, such as medical imaging or autonomous systems, where resistance to adversarial perturbations is essential.
> - **Least Promising Scenarios**:
>   Conversely, the current implementation may be less effective in scenarios where computational resources are more limited than human annotations. In these cases, the additional computational overhead introduced by the self-calibration process could limit its applicability. These limitations, however, represent opportunities for future research and optimization.
>
> We believe these considerations provide a balanced view of the proposed method's potential, outlining both its strengths and areas for future enhancement.
>
> > Classifier gets time-dependently noised/diffused samples, is trained on them with cross-entropy and score-matching loss (based on gradient of sum of logits)?
>
> You are very right. In addition to training time-dependent classifiers with perturbed (time-dependent) samples and time-dependent cross-entropy loss, we propose to use the denoising score-matching loss as a regularizer. We will include a more detailed introduction of our proposed method to clarify its formulation.
>
> > I think Figure 2 and caption explain this fairly well, but saw it quite late, maybe a similar text could be integrated somewhere in the main text as well, or figure 2 maybe shifted more towards section 3, so it can be seen alongside more easily?
>
> We appreciate your suggestion to improve our presentation. We will move Figure 2 to page 5 (the first page of Section 3) in the revision.

---

### Review · Reviewer_i9mQ · 2025-02-07

**Summary Of Contributions:**

This paper introduces Semi-Supervised Classifier Guidance with Self-Calibration (CGSGM-SC), a novel approach for conditional score-based generative models (SGMs). It addresses the overfitting of classifiers in classifier-guided SGMs (CGSGMs), particularly when trained with limited labeled data. The authors propose a self-calibration mechanism inspired by energy-based models (EBMs), allowing the classifier to regulate itself by interpreting it as an unconditional SGM. The self-calibration loss is applied to both labeled and unlabeled data, improving conditional generation quality while leveraging unlabeled data for improved performance.

Experiments on CIFAR-10 and CIFAR-100 datasets demonstrate that CGSGM-SC outperforms existing CGSGMs and classifier-free SGMs (CFSGMs), especially in semi-supervised settings. The results indicate significant improvements in conditional generation quality (e.g., 38.4 -> 7.26 in FID compared to JEM), reduced classifier overfitting, and enhanced robustness against adversarial attacks. The study further validates that CGSGM-SC successfully mitigates the drawbacks of prior CGSGM approaches while maintaining computational efficiency compared to other advanced guidance techniques. These findings reinforce the practical importance of self-calibration for semi-supervised generative models, particularly when labeled data is scarce.

**Audience:**

Yes

**Claims And Evidence:**

Yes

**Requested Changes:**

1. Provide a deeper theoretical analysis explaining how self-calibration loss enhances classifier regularization and aligns with the underlying data distribution. A formal proof or theoretical justification would significantly bolster the paper’s credibility.

2. Include runtime benchmarks comparing CGSGM-SC to vanilla CGSGM and CG-DLSM across different batch sizes and datasets. A direct comparison of computational efficiency would help practitioners assess its real-world feasibility.

3. Compare CGSGM-SC against more CFSGMs, particularly strong CFG variants. Justify why CGSGMs remain relevant in semi-supervised settings, particularly when CFSGMs have seen success in recent work. Also, compare with CFG, and justify why CFG does not perform well in semi-supervised setup.

4. Provide more details on the adversarial purification process and compare it with standard adversarial training baselines. A discussion on the effectiveness of self-calibration in adversarial robustness compared to established adversarial training methods would be beneficial.

5. Expand on how CGSGM-SC scales to high-resolution images and its potential deployment challenges. Testing on more complex datasets, such as ImageNet, and discussing memory and efficiency considerations would provide a clearer picture of its practicality in real-world applications.

**Strengths And Weaknesses:**

### Strengths
1. The paper introduces a well-motivated and theoretically sound extension of CGSGMs by leveraging self-calibration from EBMs, especially useful in semi-supervised generative modeling. Unlike prior CGSGM, CGSGM-SC successfully incorporates unlabeled data, making it more applicable to real-world scenarios with limited labeled data. The experimental results convincingly show that CGSGM-SC surpasses previous methods, including CG-DLSM and classifier-free SGMs, in terms of conditional generation quality and diversity.

2. The paper highlights an additional advantage of self-calibration—enhanced robustness against adversarial attacks, which increases its practical applicability. The study includes well-structured qualitative and quantitative evaluations, such as FID, coverage, density metrics, and ablation studies, reinforcing the validity of its claims. Additionally, the authors present extensive ablation studies, demonstrating the contribution of each component of their method, particularly the impact of self-calibration loss on classifier regularization. These results make a strong case for CGSGM-SC as a viable and scalable approach in semi-supervised learning.

3. The approach is designed to be computationally efficient and does not require the costly MCMC sampling techniques used in training energy-based models with contrastive divergence. This contributes to its practicality in large-scale generative modeling applications. The authors also provide a thorough comparison with existing CGSGMs, ensuring their claims are well-supported with empirical evidence across multiple datasets and settings.
---
### Weakness
1. While, the paper draws inspiration from EBMs and JEM, it lacks a rigorous mathematical derivation explaining why self-calibration improves classifier guidance. A more detailed theoretical analysis of CGSGM-SC’s convergence properties would enhance its contribution. Specifically, if the author can provide a formal proof of why or how self-calibration leads to better generative modeling would strenghthen the contribution of the paper.

2. The study primarily compares CGSGMs but lacks sufficient comparison on classifier-free guidance (CFG) methods. Specifically, is there any reason why CFG performs worse in semi-supervised settings? If any, a clearer justification for CGSGMs' superiority in semi-supervised settings is necessary. More discussion on scenarios where CGSGMs outperform CFSGMs in realistic conditions would be beneficial.

3. While the proposed method relies on score-based networks (e.g., NCSN), showing the applicability with diffusion models (or flow matching models) who are the SOTA generative models, would clearly improve the quality of the paper. Especially, this would allow to use advanced sampler (e.g., DDIM) instead of Predictor-Corrector sampling step, and the generation quality would be improved as well.

4. While the adversarial purification results are intriguing, the methodology lacks clarity. Are the purified images visually inspected? How does CGSGM-SC compare to other adversarial training techniques? More rigorous adversarial evaluations (e.g., AutoAttack benchmarks) would improve credibility. The study focuses on CIFAR-10 and CIFAR-100 but lacks discussion on real-world applications. Would CGSGM-SC generalize well to high-resolution datasets like ImageNet? A discussion on scalability and deployment considerations would be valuable.

5. Additionally, the paper could benefit from an explicit discussion on potential failure cases or limitations of CGSGM-SC. Understanding its weaknesses in different experimental conditions would help guide future research and practical implementations.

---

> ### Author Response · Authors · 2025-02-13
> **Response to Reviewer i9mQ by Authors Part 1**
>
> > 1. Provide a deeper theoretical analysis explaining how self-calibration loss enhances classifier regularization and aligns with the underlying data distribution. A formal proof or theoretical justification would significantly bolster the paper’s credibility.
>
> Thank you for the suggestion. While our design stems from the sound mathematical framework of EBMs and their close relationship with SGMs, we admit that we do not have a proof of how the proposed loss improves CGSGMs at the current point. Instead, we justify our claim that the proposed method indeed improves CGSGMs and is superior to other SGMs for semi-supervised conditional generation through evidence collected from empirical results and ablation studies. For instance,
> 1. In semi-supervised settings on CIFAR-10 and CIFAR-100, our method consistently outperforms CFSGMs and other CGSGM variants.
> 2. In our experiment with the toy dataset, we visually demonstrate that classifiers trained solely with the cross-entropy loss predict inaccurate gradients, whereas the incorporation of our self-calibration loss leads to more accurate gradient estimation.
>
> We believe that the detailed empirical results and ablation studies provide accurate, convincing, and clear evidence in support of our claims--well in line with TMLR's acceptance criteria. We will note in the revision that the lack of a formal theoretical proof is a current limitation and a potential future direction.
>
> > 2. Include runtime benchmarks comparing CGSGM-SC to vanilla CGSGM and CG-DLSM across different batch sizes and datasets. A direct comparison of computational efficiency would help practitioners assess its real-world feasibility.
>
> Thank you for the suggestion. We totally agree with this, and before the coming week, we will include discussions on the computational efficiency of CG, CG-DLSM, and the proposed method in the appendix of the revision.
>
> > 3. Compare CGSGM-SC against more CFSGMs, particularly strong CFG variants. Justify why CGSGMs remain relevant in semi-supervised settings, particularly when CFSGMs have seen success in recent work. Also, compare with CFG, and justify why CFG does not perform well in semi-supervised setup.
>
> Thank you for highlighting the importance of comparing against a broader range of CFG-based solutions. We first hope to clarify that the CFSGM architecture that we are using is the only one we know that has been carefully studied with CIFAR data. Other CFSGMs' success highly depends on training with larger-scale data, which cannot give us a fair comparison. We thus choose to stick with the current architecture. Within the architecture, we have attempted to consider a broad spectrum of representative variants, including
> - Cond: Training the model solely on labeled data and sampling directly from the estimated conditional score.
> - CFG-labeled: Utilizing all available labeled data to train the conditional path $s_\theta(x\vert y)$ while training the unconditional path $s_\theta(x)$ only on labeled data.
> - CFG-all: Instead of using only labeled data to train the unconditional path $s_\theta(x)$, utilize all data for training.
>
> We believe that the three representatives form sufficient ablation studies and evidence to help understand why CFSGM does not perform well on semi-supervised conditional generation. While CFSGMs are great at generating class-conditional images with high accuracy, as shown by their superior intra-Density scores, they possess significant drawbacks in semi-supervised settings. Specifically, our experimental results (Table 2 and Fig. 4) showed that CFSGMs suffer from reduced diversity with limited labeled data, as indicated by significant drops in Coverage and intra-Coverage. This drop is primarily due to their reliance on the labeled data during training, which causes the model to closely mirror the labeled distribution rather than capturing the full variety of both labeled and unlabeled data. In other words, the conditional paths of CFSGMs tend to severely overfit on labeled data, causing the sample diversity to deteriorate when labeled data is scarce.
>
> We will include relevant discussions above to clarify our choice of CFSGM architecture and to provide more insights on the deficiency of CFSGMs to stimulate future research in the revision in the coming week.

---

> > ### Author Response · Authors · 2025-02-13
> > **Response to Reviewer i9mQ by Authors Part 2**
> >
> > > 4. Provide more details on the adversarial purification process ...
> >
> > Thanks for the suggestion. We agree to that and plan to incorporate an algorithm illustrating the adversarial purification process in Section 5.
> >
> > > 4. ... compare it with standard adversarial training baselines. A discussion on the effectiveness of self-calibration in adversarial robustness compared to established adversarial training methods would be beneficial.
> >
> > Thank you. Given that we use a different architecture than the current adversarial training works and the two-week revision time constraint, we plan to implement only the most representative methods on our model architecture as a proof of concept. In particular, we plan to run AutoAttack against the proposed method. We expect to include the results in Section 5 in the coming week.
> >
> > > 5. Expand on how CGSGM-SC scales to high-resolution images and its potential deployment challenges. Testing on more complex datasets, such as ImageNet, and discussing memory and efficiency considerations would provide a clearer picture of its practicality in real-world applications.
> >
> > We agree that evaluating the method on higher-resolution datasets such as ImageNet would provide valuable insights into its real-world applicability. Unfortunately, due to current computational resource constraints in our academic environment, our experiments have been limited to lower-resolution images and smaller scale datasets. Additionally, the proposed method's reliance on unconditional SGMs has posed challenges, as we cannot find a properly-trained unconditional SGM for resolutions beyond 32x32. Thus, we are not able to complete such experiments within the two-week revision time frame. We acknowledge the scalability challenges and will discuss them as limitations and directions for future research in the revision. On the other hand, we believe that our current experiments on lower-resolution images have provided sufficient evidence in support of our claims that SC-regularized CGSGMs are promising for semi-supervised conditional generation, which matches TMLR's acceptance criteria.
> >
> > > W3. While the proposed method relies on score-based networks (e.g., NCSN), showing the applicability with diffusion models (or flow matching models) who are the SOTA generative models, would clearly improve the quality of the paper. Especially, this would allow to use advanced sampler (e.g., DDIM) instead of Predictor-Corrector sampling step, and the generation quality would be improved as well.
> >
> > We agree with the importance of applicability with SOTA generative models like diffusion models. In [1], the authors revealed that diffusion models can be reinterpreted as SGMs. Our empirical results demonstrated that the proposed method improves the scores estimated by CGSGMs. Therefore, we believe the proposed method could be extended to enhance classifier guidance in diffusion models. In the revision, we will include a discussion on extending the proposed method to diffusion models to clarify how we can integrate the self-calibration loss with these advanced models as a future direction.
> >
> > [1] Song et al. Score-based generative modeling through stochastic differential equations, ICLR 2021

---

### Author Response · Authors · 2025-02-13
**Global Response to Reviewers by Authors**

We thank all reviewers for their constructive and professional comments. In this preliminary response, we address the various concerns raised to facilitate the discussion within the limited two-week revision window. We plan to edit our paper and upload the revised version in the coming week. We appreciate your insightful feedback, which has greatly contributed to improving our work, and we look forward to further constructive discussion.

---

### Decision · Action_Editor_DrX3 · 2025-03-28

**Recommendation:** Reject

**Comment:**

The paper introduces a self-calibration loss to improve classifier guidance in semi-supervised score-based generative models. While the idea is supported by experiments on CIFAR-10/100, it lacks theoretical justification (Reviewer i9mQ) and broader experimental validation (Reviewer i9mQ, Reviewer tVur). Key comparisons to alternative methods, such as VAEs, GANs, and diffusion models with pseudo-labeling, are missing (Reviewer tVur). The robustness results are underdeveloped and lack rigorous benchmarks (Reviewer tVur, Reviewer wDV4). Despite thoughtful responses, the paper does not sufficiently address these concerns around generalizability, theoretical clarity, and empirical scope. At this stage, the submission does not satisfy the criteria for publication.

**Audience:**

Although the topic may interest researchers working on classifier-guided generative models, its relevance to the broader TMLR audience is limited by its narrow scope. Additional evaluations on more complex datasets or stronger theoretical insights will be interested to broader ML community.

**Claims And Evidence:**

The paper proposes a self-calibration loss to improve classifier guidance in semi-supervised score-based generative models. While empirical results on CIFAR-10/100 demonstrate improvements in intra-FID and robustness, the evidence lacks theoretical depth and broader validation. There is no formal justification for why the proposed loss improves classifier gradients, and the experiments are limited to small-scale datasets with minimal comparison to stronger semi-supervised baselines such as GANs, VAEs, or advanced diffusion models. The adversarial robustness claim, although promising, is under-explored and lacks comparisons with standard defenses, further limiting the strength of the evidence.

**Resubmission Of Major Revision:**

The authors may consider submitting a major revision at a later time.